# Accelerating Batch Active Learning Using Continual Learning Techniques

**Arnav Das**[* †]   **Gantavya Bhatt**[*†]   **Megh Bhalerao**[†]   **Vianne Gao**[◇]   **Rui Yang**[◇]   **Jeff Bilmes**[†]
[†]**University of Washington, Seattle**     [◇] **Memorial Sloan Kettering Cancer Center**
`{arnavmd2, gbhatt2, bilmes}@uw.edu`

**Reviewed on OpenReview:** `https://openreview.net/forum?id=T55dLSgsEf`

## Abstract

A major problem with Active Learning (AL) is high training costs since models are typically retrained from scratch after every query round. We start by demonstrating that standard AL on neural networks with warm starting fails, both to accelerate training and to avoid catastrophic forgetting when using fine-tuning over AL query rounds. We then develop a new class of techniques, circumventing this problem, by biasing further training towards previously labeled sets. We accomplish this by employing existing, and developing novel, replay-based Continual Learning (CL) algorithms that are effective at quickly learning the new without forgetting the old, especially when data comes from an evolving distribution. We call this paradigm *"Continual Active Learning" (CAL)*. We show CAL achieves significant speedups using a plethora of replay schemes that use model distillation and that select diverse/uncertain points from the history. We conduct experiments across many data domains, including natural language, vision, medical imaging, and computational biology, each with different neural architectures and dataset sizes. CAL consistently provides a $\sim$3x reduction in training time, while retaining performance and out-of-distribution robustness, showing its wide applicability.

## 1  Introduction

While neural networks have been successful in a variety of different supervised settings, most such approaches are labeled-data hungry and require significant computation. From a large pool of unlabeled data, active learning (AL) selects subsets of points to label by imparting the learner with the ability to query a human annotator. Such methods incrementally add points to the labeled pool by repeatedly: (1) training a model from scratch on the current labeled pool and (2) using some measure of model uncertainty and/or diversity to select a set of points to query the annotator (Settles, 2009; 2011; Wei et al., 2015; Ash et al., 2020; Killamsetty et al., 2021a). AL has been shown to reduce the amount of training data required but can be computationally expensive since it requires retraining a model, typically from scratch, after each query round.

A *simple* solution is to warm start the model parameters between query rounds. However, the observed speedups tend to still be limited since the model must make several passes through an ever-increasing pool of data. Moreover, warm starting alone in some cases can hurt generalization, as discussed in Ash & Adams (2020) and Beck et al. (2021). Another extension to this is to solely train on the newly labeled batch of examples to avoid re-initialization. However, as we show in Section 4, naive fine-tuning fails to retain accuracy on previously seen examples since the distribution of the query pool may drastically change with each round.

This problem of *catastrophic forgetting* while incrementally learning from a series of new tasks with shifting distributions is a central question in another paradigm called Continual Learning (CL) (French, 1999; McCloskey & Cohen, 1989; McClelland et al., 1995; Kirkpatrick et al., 2017c). CL has recently gained popularity, and many algorithms have been introduced to allow models to quickly adapt to new tasks without

---

*Equal contribution

forgetting (Riemer et al., 2018; Lopez-Paz & Ranzato, 2017; Chaudhry et al., 2019; Aljundi et al., 2019b; Chaudhry et al., 2020; Kirkpatrick et al., 2017b).

In this work, we propose Continual Active Learning (CAL)[1], which applies Continual Learning strategies to accelerate batch Active Learning. In CAL, we apply CL to enable the model to learn the newly labeled points without forgetting previously labeled points while using past samples efficiently using *replay-based* methods. As such, we observe that CAL attains significant training time speedups over standard AL. This is beneficial for the following reasons: **(1):** As neural networks swell (Shoeybi et al., 2019), so do the environmental and financial model training costs (Bender et al., 2021; Dhar, 2020; Schwartz et al., 2020). Reducing the number of gradient updates required for AL will help mitigate such costs, especially with large-scale models. **(2):** Reducing AL computational requirements makes AL more accessible for edge computing, IoT, and low-resource device deployment (Senzaki & Hamelain, 2021) such as with federated learning (Li et al., 2020). **(3):** Developing new AL algorithms/acquisition functions, or searching for architectures as done with NAS/AutoML that are well-suited *specifically* for AL, can require hundreds or even thousands of runs. Since CAL's speedups are agnostic to the AL algorithm and the neural architecture, such experiments can be significantly sped up. Overall, the importance of speeding machine learning training processes is well recognized, as evidenced by the plethora of efforts in the computing systems community (Jia et al., 2022; Zhang et al., 2017; Zheng et al., 2022).

In addition, CAL demonstrates another practical application for CL methods. Many settings used to benchmark CL methods in recent work are somewhat contrived (Farquhar & Gal, 2018; Wallingford et al., 2023). Most CL work considers the class/domain incremental setting, where only the samples that belong to a subset of the set of classes/domains of the original dataset are available to the model at any given time. This setting need not be the only benchmark upon which CL methods are evaluated. We suggest that the evaluation of future new CL algorithms should be determined not only on traditional CL evaluation schemes and benchmarks but also on their performance in the CAL setting.

In the present work, we are not using CL to improve or change AL querying strategies. We view this as both a strength of the present work and an opportunity for future work. Firstly, it is a strength of our present work since any AL query strategy, both old and new, can in principle be applied in the CAL

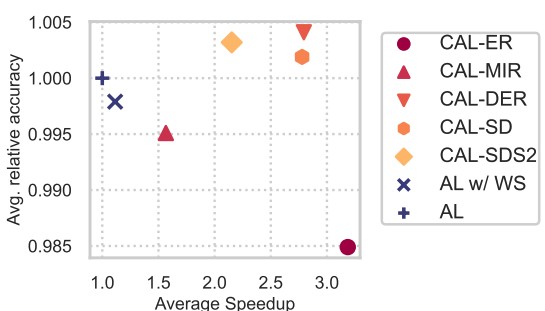

Figure 1: Summary of our results: Benchmarking average relative accuracy vs. speedup for the proposed methods (*solid markers*) against the baseline. Each point is averaged across datasets and labeling budgets; *top-right region is desiderata*. This shows that proposed CAL methods (particularly CAL-DER/SD/SDS2) have equivalent performance relative to the baseline (if not better) on multiple datasets and across different budgets, all while providing significant speedups.

setting, both for speeding up the AL strategy and offering, as argued above, a test bed for CL. Indeed, a major AL research challenge includes combining AL with other techniques, as we believe we have done herein. Secondly, it is an opportunity since there is nothing inherent in the definition of CL that precludes the CL tasks from being dependent on the model as we show below. There may be a way for CL to open doors to intrinsically new AL querying policies of attaining new batches of unlabeled data. This we leave to the future. Our present *core goal is to accelerate batch AL training via CL techniques while preserving accuracy.*

To the best of our knowledge, this application of CL algorithms to accelerate batch AL has never been explored. Our contributions can be summarized as follows: **(1)** We first demonstrate that batch active learning techniques can benefit from continual learning techniques, and their merger creates a new class of techniques that we propose to call the "CAL framework." **(2)** We benchmark several existing CL methods (CAL-ER, CAL-DER, CAL-MIR) as well as novel methods (CAL-SD, CAL-SDS2) and evaluate them on diverse datasets based on the accuracy/speedup they can attain over standard AL. **(3)** We study speedup/performance trade-offs on datasets that vary in modality (natural language, vision, medical imaging, and computational biology), neural architecture with varying degrees of computation (Transformers/CNNs/MLPs), data scale (including

---

[1]There have been papers published with titles containing the "Continual Active Learning" phrase, but these do not merge **Continual** Learning with **Active Learning** as we do, hence our name.

some larger datasets, one having 2M samples), and class-balance. And **(4)**, lastly, we demonstrate that models trained with CAL and standard AL models behave similarly, in that both classes of models attain similar uncertainty scores on held-out datasets and achieve similar robustness performance on out-of-distribution data. Figure 1 summarizes our results, detailed later in the paper and greatly detailed in the appendices.

## 2 Related Work

Active learning (Atlas et al., 1989; Cohn et al., 1994; Wei et al., 2015; Killamsetty et al., 2021a; Ash et al., 2020) has demonstrated label efficiency over passive learning. In addition, there has been extensive work on theoretical aspects of AL (Guillory et al., 2009; Hanneke, 2009; 2007; Balcan et al., 2010) where Hanneke (2012) shows sample complexity advantages over passive learning in noise-free classifier learning for VC classes. More recently, Active Learning has also been studied as a procedure to incrementally learn the underlying data distribution with the help of discrepancy framework (Mathelin et al., 2022; Cui & Sato, 2020; Shui et al., 2019).

Recently there has been an interest in speeding up active learning since most deep learning is computationally demanding. Kirsch et al. (2019); Pinsler et al. (2019); Sener & Savarese (2018) aim to reduce the number of query iterations by having large query batch sizes. However, they do not exploit the learned models from previous rounds for the subsequent ones and are therefore complementary to CAL. Work such as Coleman et al. (2020a); Ertekin et al. (2007); Mayer & Timofte (2020); Zhu & Bento (2017); Zhang et al. (2023) speeds up the selection of the new query set by appropriately restricting the search space or by using generative methods. This work can be easily integrated into our framework because CAL works on the training side of active learning, not on the query selection. On the other hand, Lewis & Catlett (1994); Coleman et al. (2020b); Yoo & Kweon (2019) use a smaller proxy model to reduce computation overhead, however, they still follow the standard active learning protocol, and therefore can be accelerated when integrated with CAL.

There exists work that explores continual/transfer learning and active learning in the same context. Perkonigg et al. (2021) propose an approach that allows active learning to be applied to data streams of medical images by introducing a module that detects domain shifts. This is quite distinct from our work: our work uses CL algorithms to prevent catastrophic forgetting and to accelerate learning. Zhou et al. (2021) study when standard active learning is used to fine-tune a pre-trained model, and employs transfer learning — this does not consider continual learning and active learning together, however, and is therefore not related to our work. Finally, Ayub & Fendley (2022) studies where a robot observes unlabeled data sampled from a shifting distribution, but does not explore active learning acceleration.

For preventing catastrophic forgetting, we mostly focus on replay-based algorithms that are state-of-the-art methods in CL. However, as demonstrated in Section 4 on how active learning rounds can be viewed in a continual learning context, one can apply other methods such as EWC (Kirkpatrick et al., 2017a), Bayesian divergence priors Li & Bilmes (2007), structural regularization (Li et al., 2021) or functional regularization (Titsias et al., 2020) as well.

The effect of warm-started model training on generalization and convergence speed has been explored by Ash & Adams (2020) which empirically demonstrates that a model that has been pretrained on a source dataset converges faster but exhibits worse generalization on a target dataset when compared to a randomly initialized model. However, that work only considers the setting where the source and target datasets are unbiased estimates of the same distribution. This is distinct from our work since the distributions we consider are all dependent on the model at each AL round. Furthermore, our work employs CL methods in addition to warm-starting, also not considered in Ash & Adams (2020).

## 3 Background

### 3.1 Batch Active Learning

Define $[n] = \{1, ..., n\}$, and let $\mathcal{X}$ and $\mathcal{Y}$ denote the input and output domains respectively. AL typically starts with an unlabeled dataset $\mathcal{U} = \{x_i\}_{i \in [n]}$, where each $x_i \in \mathcal{X}$. The AL setting allows the model $f$, with parameters $\theta$, to query a user for labels for any $x \in \mathcal{U}$, but the total number of labels is limited to a budget $b$, where $b \leq n$. Throughout the work, we consider classification tasks so the output of $f(x; \theta)$ is a probability

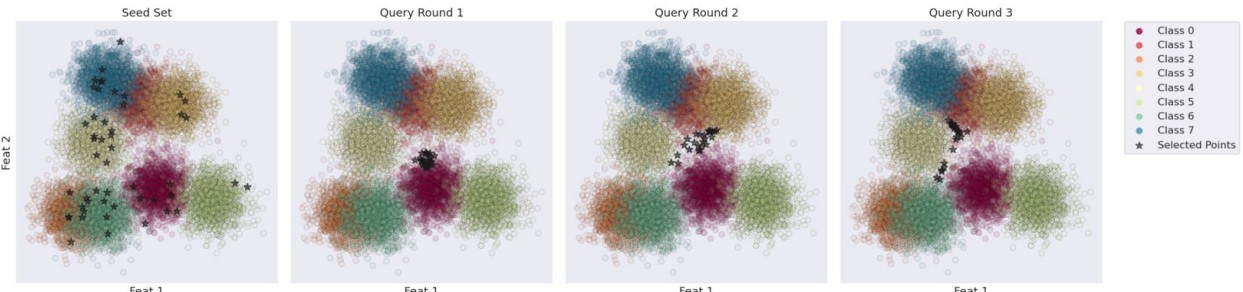

Figure 2: Visualization of standard AL when training a linear model on a simple 2D synthetic dataset. Each plot shows which samples were chosen by uncertainty sampling to be added to the labeled pool at given a query round. Notably, the distributions of the new labeled points vary significantly between rounds motivating CAL.

distribution over classes. The goal of AL is to ensure that $f$ can attain low error when trained only on the set of $b$ labeled points.

Algorithm 1 details the general AL procedure. Lines 3-6 construct the seed set $\mathcal{D}_1$ by randomly sampling a subset of points from $\mathcal{U}$ and labeling them. Lines 7-14 iteratively expand the labeled set for $T$ rounds by training the model from a random initialization on $\mathcal{D}_t$ until convergence and selecting $b_t$ points (where $\sum_{t \in [T]} b_t = b$) from $\mathcal{U}$ based on some selection criteria that is dependent on $\theta_t$. The selection criteria generally select samples based on model uncertainty and/or diversity (Lewis & Gale, 1994; Dagan & Engelson, 1995; Settles; Killamsetty et al., 2021a; Wei et al., 2015; Ash et al., 2020; Sener & Savarese, 2017). In this work, we primarily consider uncertainty sampling Lewis & Gale (1994); Dagan & Engelson (1995); Settles, though we also test other selection criteria in Section C in the Appendix.

**Uncertainty Sampling** is a widely-used practical AL method that selects $\mathcal{U}_t = \{x_1, \ldots, x_{b_t}\} \subseteq \mathcal{U}$ by choosing the samples that maximize a notion of model uncertainty. We consider entropy (Dagan & Engelson, 1995) as the uncertainty metric, so if $h_\theta(x) \triangleq -\sum_{i \in [k]} f(x;\theta)_i \log f(x;\theta)_i$, then $\mathcal{U}_{t+1} \in \underset{\mathcal{A} \subset \mathcal{U}:|\mathcal{A}|=b_t}{\operatorname{argmax}} \sum_{x \in \mathcal{A}} h_{\theta_t}(x)$.

**Distribution Shift** As shown in Algorithm 1, the set of samples that are labeled at a given query round are dependent on the model parameters. Since the model is

---

**Algorithm 1** Batch Active Learning

1: **procedure** ACTIVELEARNING($f$, $\mathcal{U}$, $b_{1:T}$, $T$)
2:     $t \leftarrow 1, \mathfrak{L} \leftarrow \emptyset$              ▷ Initialize
3:     $\mathcal{U}_t \sim \mathcal{U}$              ▷ Draw $b_1$ samples from $\mathcal{U}$
4:     $\mathcal{D}_t \leftarrow \{(x_i, y_i)|x_i \in \mathcal{U}_t\}$     ▷ Provide labels
5:     $\mathcal{U} \leftarrow \mathcal{U} \setminus \mathcal{U}_t$        ▷ Remove from unlabeled set
6:     $\mathfrak{L} \leftarrow \mathfrak{L} \cup \mathcal{D}_t$           ▷ Add to labeled set
7:     **while** $t \leq T$ **do**
8:         Randomly initialize $\theta_{init}$
9:         $\theta_t \leftarrow \text{Train}(f, \theta_{init}, \mathfrak{L})$
10:       $\mathcal{U}_{t+1} \leftarrow \text{Select}(f, \theta_t, \mathcal{U}, b_t)$   ▷ Select $b_t$ points based on $\theta_t$
11:       $\mathcal{D}_{t+1} \leftarrow \{(x_i, y_i)|x_i \in \mathcal{U}_t\}$
12:       $\mathcal{U} \leftarrow \mathcal{U} \setminus \mathcal{U}_{t+1}; \mathfrak{L} \leftarrow \mathfrak{L} \cup \mathcal{D}_{t+1}; t \leftarrow t+1$
13:     $\theta_T \leftarrow \text{Train}(f, \theta_{init}, \mathfrak{L})$
14:     **return** $\mathfrak{L}, \theta_T$

---

repeatedly updated, the distribution of newly labeled samples may vary across different AL rounds. This is illustrated in Figure 2, where we perform standard AL in a simplified setting. In the example, we use a linear model to perform classification on a 2D synthetic dataset and perform AL with entropy-based uncertainty sampling as the acquisition function. Upon visualizing the set of samples that are queried by the model at different AL rounds, it is evident that distribution shift does occur. We demonstrate empirically in Figure 3 that training on newly labeled samples alone will cause catastrophic forgetting due to distribution shifts.

## 3.2 Continual Learning

We define $\mathcal{D}_{1:n} = \bigcup_{i \in [n]} \mathcal{D}_i$. In CL, the dataset consists of $T$ tasks $\{\mathcal{D}_1, ..., \mathcal{D}_T\}$ that are presented to the model sequentially, where $\mathcal{D}_t = \{(x_i, y_i)\}_{i \in N_t}$, $N_t$ are the task-$t$ sample indices, and $n_t = |N_t|$. At time $t \in [T]$, the data/label pairs are sampled from the current task $(x, y) \sim \mathcal{D}_t$, and the model has only limited access to the history $\mathcal{D}_{1:t-1}$. The CL objective is to efficiently adapt the model to $\mathcal{D}_t$ while ensuring performance on the history does not appreciably degrade. We focus on replay-based CL techniques that attempt to

| **Continual Learning Papers** | **History Access** | | |
| --- | --- | --- | --- |
| | **None** | **Partial** | **Full** |
| Kirkpatrick et al. (2017b), Li & Hoiem (2017) | ✓ | × | × |
| Aljundi et al. (2019b), Mai et al. (2020) | × | ✓ | × |
| Buzzega et al. (2020), Aljundi et al. (2019a) Lopez-Paz & Ranzato (2017), Chaudhry et al. (2019) Ratcliff (1990) | × | ✓ | ✓ |

Table 1: Different previous CL work organized by the memory access assumptions made in their settings. In the CAL setting, we do not need to assume that we have limited access to the history so we only consider CL methods that are applicable to the setting where the history is fully accessible. This category primarily contains replay-based methods.

approximately solve CL optimization by using samples from $\mathcal{D}_{1:t-1}$ to regularize the model while adapting to $\mathcal{D}_t$. Please refer to appendix B for more details on CL.

Algorithm 2 outlines general replay-based CL, where the objective is to adapt $f$ parameterized by $\theta_0$ to $\mathcal{D}$ while using samples from the history $\mathcal{H}$. $\mathcal{B}_{\text{current}}$ consists of $m$ points randomly sampled from the *current time's* $\mathcal{D}$, and $\mathcal{B}_{\text{replay}}$ consists of $m^{(h)}$ points chosen based on a criterion that selects from $\mathcal{H}$. In line 6, $\theta_\tau$ is computed based on an update rule that utilizes both $\mathcal{B}_{\text{replay}}$ and $\mathcal{B}_{\text{current}}$.

---
**Algorithm 2** Continual Learning
---
1: **procedure** CONTINUALTRAIN($f, \theta_0, \mathcal{D}, \mathcal{H}, m, m^{(h)}$)
2:     $\tau \leftarrow 0$          ▷ $\tau$ is a local iteration index.
3:     **while** not converged **do**
4:         $\tau \leftarrow \tau + 1$
5:         $\mathcal{B}_{\text{current}} \leftarrow \{(x_i, y_i)\}_{i=1}^{m} \sim \mathcal{D}$   ▷ Sample from new task.
6:         $\mathcal{B}_{\text{replay}} \leftarrow \text{Select}(f, \theta_{\tau-1}, \mathcal{H}, m^{(h)})$   ▷ Sample from history.
7:         $\theta_\tau \leftarrow \text{Update}(f, \theta_{\tau-1}, \mathcal{B}_{\text{current}}, \mathcal{B}_{\text{replay}})$
8:     return $\theta_\tau$
---

Different CL methods assume varying degrees of access to the history. In many practical settings, $\mathcal{D}_{1:T}$ is too large to store in memory or $T$ is unknown so CL algorithms assume limited or no access to samples from the history. Some of these approaches remove unimportant samples from the history Aljundi et al. (2019b); Mai et al. (2020), while others solely rely on regularizing the model parameters without replaying any samples from the history Kirkpatrick et al. (2017b); Li & Hoiem (2017). In the CAL setting, $\mathcal{D}_{1:T}$ is already stored in memory as is done in the standard AL setting so employing a CL algorithm that imposes a memory constraint would needlessly cause CAL to underperform. In Table 1, a handful of prior CL works are sorted by what assumptions about memory/history access they make; methods that do not make any memory constraint assumption are the most well-suited for CAL. A more comprehensive and complete overview of CL methods can be found in De Lange et al. (2022).

## 4 Blending Continual and Active Learning

A clear AL inefficiency is that the model $f$ is retrained from scratch on the entire labeled pool after every query round. One potential solution idea is to simply continue training the model only on the newly AL-queried samples and, via the process of warm starting, hope that history will not fade. Unfortunately for this approach, Figure 3 (top) shows that when the model is warm-start trained *only* on the task $t$ (samples labeled at AL round $t$ using entropy sampling), historical sample performance deteriorates precipitously while performance on the validation set flatlines. That is, at AL round $t$ (x-axis), we continue to train the model until convergence on task $t$ and track accuracy (y-axis) on each previous task and also on the validation set. The performance of task 1, after the initial drop, tracks that of the validation set, since task 1 is a model agnostic initial unbiased random subset query of the training data. The performance of task $i$, for $i > 1$, however, each of which is the result of model-conditioned AL query, shows perilous historical forgetting. In the end, the model performs considerably worse on all of the historical tasks (aside from task 1) than on the validation set, even though it has been trained on those tasks and not on the validation set. This experiment suggests that: (1) the distribution of each AL-queried task $t > 1$ is different than the data distribution; (2) fine-tuning to task $t$ can result in catastrophic forgetting; and (3) techniques to combat catastrophic forgetting are necessary to effectively incorporate new information between successive AL rounds.

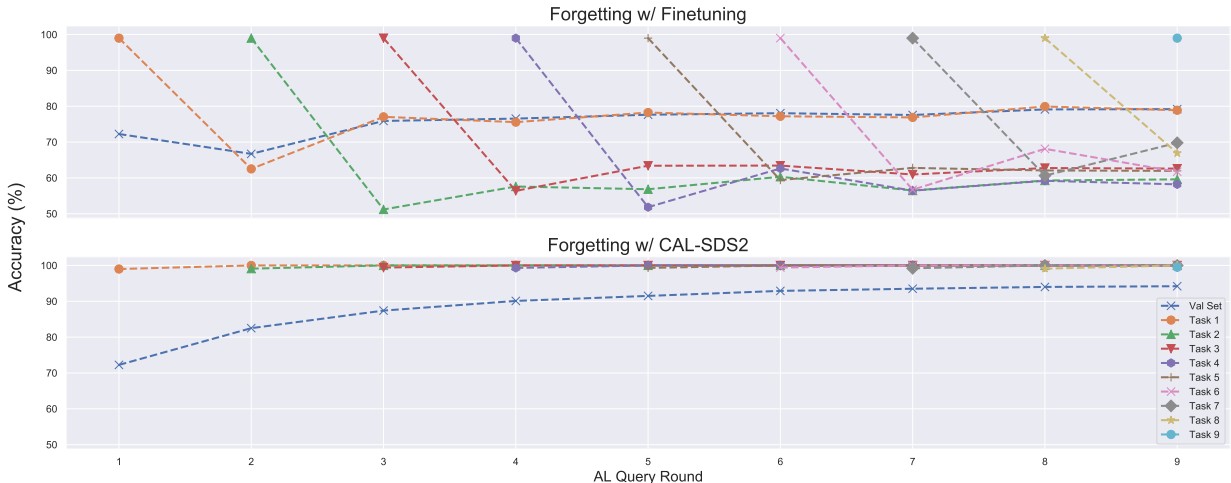

Figure 3: A ResNet-18's performance on CIFAR-10 on the validation set (blue X) and task $i$ for $i \in \{1, 2, \ldots, 9\}$ (other colors) after training the warm-started model and training just on the new task data (top) compared to efficiently adapting to newly labeled data with a CAL method (bottom). In the top plot, at each round a new 5% of the full dataset is added to the labeled pool and the previously trained model is further trained *just* on the new points (the isolated upper-right cyan point shows task 9's performance after round 9). After each round, the accuracy on the corresponding task does well, but all previous tasks' accuracies drop precipitously, demonstrating a form of catastrophic forgetting. In the bottom plot, the same setting is considered but the model is adapted to the new data with CAL-SDS2, one of the CAL methods. Also, the validation set performance has flatlined despite continued training on new data when finetuning, but continues to increase when CAL-SDS2 is employed. This demonstrates that naively fine-tuning a model to newly labeled points can work poorly.

The CAL approach, shown in Algorithm 3, uses CL techniques to ameliorate catastrophic forgetting. The key difference between CAL and Algorithm 1 is line 9. Instead of standard training, replay-based CL is used to adapt $f$ to $\mathcal{D}_t$ while retaining performance on $\mathcal{D}_{1:t-1}$. The speedup comes from two sources: (1) we are computing gradient updates only on a useful subset of the history $\mathcal{D}_{1:t-1}$ rather than all of it for reasonable choices of $m^{(h)}$; and (2) the model converges faster since it starts warm. The ability of CAL to combat catastrophic forgetting is shown in Figure Figure 3 (bottom). In the rest of the section, define

**Algorithm 3** The general CAL approach.

1: **procedure** CAL($f$, $\mathcal{U}$, $b_{1:T}$, $T$, $m$, $m^{(h)}$)
2:     $t \leftarrow 1$, $\mathfrak{L} \leftarrow \emptyset$         ▷ Initialize
3:     $\mathcal{U}_t \sim \mathcal{U}$         ▷ Draw $b_1$ samples from $\mathcal{U}$
4:     $\mathcal{D}_t \leftarrow \{(x_i, y_i)|x_i \in \mathcal{U}_t\}$         ▷ Provide labels
5:     $\mathcal{U} \leftarrow \mathcal{U} \setminus \mathcal{U}_t$
6:     $\mathfrak{L} \leftarrow \mathfrak{L} \cup \mathcal{D}_t$
7:     Randomly initialize $\theta_0$
8:     **while** $t \leq T$ **do**
9:        $\theta_t \leftarrow \text{ContinualTrain}(f, \theta_{t-1}, \mathcal{D}_t, \mathcal{D}_{1:t-1}, m, m^{(h)})$
10:       $\mathcal{U}_{t+1} \leftarrow \text{Select}(f, \theta_t, \mathcal{U}, b_t)$    ▷ Select $b_t$ points from $\mathcal{U}$
11:       $\mathcal{D}_{t+1} \leftarrow \{(x_i, y_i)|x_i \in \mathcal{U}_{t+1}\}$
12:       $\mathcal{U} \leftarrow \mathcal{U} \setminus \mathcal{U}_{t+1}$; $\mathfrak{L} \leftarrow \mathfrak{L} \cup \mathcal{D}_{t+1}$; $t \leftarrow t + 1$
13:     $\theta_T \leftarrow \text{ContinualTrain}(f, \theta_{T-1}, \mathcal{D}_T, \mathcal{D}_{1:T-1}, m, m^{(h)})$
14:     **return** $\mathfrak{L}$, $\theta_T$

$\mathcal{L}_c(\theta) \triangleq \mathbb{E}_{(x,y) \sim \mathcal{B}_{\text{current}}}[\ell(y, f(x; \theta))]$. We next define and compare several CAL methods and assess their performance based on their performance on the test set and the speedup they attain compared to standard AL.

**Experience Replay (CAL-ER)** is the simplest and oldest replay-based method (Ratcliff, 1990; Robins, 1995). In this approach, $\mathcal{B}_{\text{current}}$ and $\mathcal{B}_{\text{replay}}$ are interleaved to create a minibatch $\mathcal{B}$ of size $m + m^{(h)}$ and $\mathcal{B}_{\text{replay}}$ is chosen uniformly at random from $\mathcal{D}_{1:t-1}$. The parameters $\theta$ of model $f$ are updated based on the gradient computed on $\mathcal{B}$.

**Maximally Interfered Retrieval (CAL-MIR)** chooses a size-$m^{(h)}$ subset of points from $\mathcal{D}_{1:t-1}$ most likely to be forgotten (Aljundi et al., 2019a). Given a batch of $m$ labeled samples $\mathcal{B}_{\text{current}}$ sampled from $\mathcal{D}_t$ and model parameters $\theta$, $\theta_v$ is computed by taking a "virtual" gradient step i.e., $\theta_v = \theta - \eta \nabla \mathcal{L}_c(\theta)$ where

$\eta$ is the learning rate. Then for every example $x$ in the history, $s_{MIR}(x) = \ell(f(x;\theta), y) - \ell(f(x;\theta_v), y)$ (i.e., the change in loss after taking a single gradient step) is computed. The $m^{(h)}$ samples with the highest $s_{MIR}$ score are selected for $\mathcal{B}_{\text{replay}}$, and the remainder is similar to experience replay. $\mathcal{B}_{\text{current}}$ and $\mathcal{B}_{\text{replay}}$ are concatenated to form the minibatch (as in CAL-ER), upon which the gradient update is computed. In practice, selection is done on a random subset of $\mathcal{D}_{1:t-1}$ for speed, since computing $s_{MIR}$ for every historical sample is prohibitively expensive.

**Dark Experience Replay (CAL-DER)** uses a distillation approach to regularize updates (Buzzega et al., 2020). Let $g(x;\theta)$ denote the pre-softmax logits of classifier $f(x;\theta)$, i.e., $f(x;\theta) = \text{softmax}(g(x;\theta))$. In DER, every $x' \in \mathcal{D}_{1:t-1}$ has an associated $z'$ which corresponds to the model's logits at the end of the task when $x$ was first observed — if $x' \in \mathcal{D}_{t'}$, then $z' \triangleq g(x';\theta_{t'}^*))$ where $t' \in [t-1]$ and $\theta_{t'}^*$ are the parameters obtained after round $t'$. DER minimizes $\mathcal{L}_{\text{DER}}(\theta)$ defined as:

$$\mathcal{L}_{\text{DER}}(\theta) \triangleq \mathcal{L}_c(\theta) + \mathop{\mathbb{E}}_{(x',y',z')\sim\mathcal{B}_{\text{replay}}} \left[ \alpha \, \|g(x';\theta) - z'\|_2^2 + \beta \, \ell(y', f(x';\theta)) \right], \tag{1}$$

where $\mathcal{B}_{\text{replay}}$ is a batch uniformly at randomly (w/o replacement) sampled from $\mathcal{D}_{1:t-1}$, and $\alpha$ and $\beta$ are tuneable hyperparameters. The first term ensures that samples from the current task are classified correctly. The second term consists of a classification loss and a mean squared error (MSE) based distillation loss applied to historical samples.

**Scaled Distillation (CAL-SD)** $\mathcal{L}_{\text{SD}}(\theta)$ is a new objective proposed in this work defined via:

$$\mathcal{L}_{\text{replay}}(\theta) \triangleq \mathop{\mathbb{E}}_{(x',y',z')\sim\mathcal{B}_{\text{replay}}} \left[ \alpha \, D_{\text{KL}}\Big(\text{softmax}(z') \,||\, f(x';\theta)\Big) + (1-\alpha) \, \ell(y', f(x';\theta)) \right], \tag{2}$$

and then $\mathcal{L}_{\text{SD}}(\theta) \triangleq \lambda_t \, \mathcal{L}_c(\theta) + (1-\lambda_t) \, \mathcal{L}_{\text{replay}}(\theta)$ where $\lambda_t \triangleq |\mathcal{D}_t|/(|\mathcal{D}_t|+|D_{1:t-1}|)$. Similar to CAL-DER, $\mathcal{L}_{\text{replay}}$ is a sum of two terms: a distillation loss and a classification loss. The distillation loss expresses the KL-divergence between the posterior probabilities produced by $f$ and $\text{softmax}(z')$, where $z'$ is defined in the DER section. We use KL-divergence instead of MSE loss on the logits so that the distillation and the classification losses have the same scale and dynamic range, and also since it allows the logits to drift by a constant term that does not affect the softmax output but does effect the MSE loss. $\alpha \in [0, 1]$ is a tuneable hyperparameter.

The weight of each term is determined adaptively by a "stability/plasticity" trade-off term $\lambda_t$. A stability-plasticity dilemma is commonly found in both biological and artificial neural networks (Abraham & Robins, 2005; Mermillod et al., 2013). A network is *stable* if it can effectively retain past information but cannot adapt to new tasks efficiently, whereas a network that is *plastic* can quickly learn new tasks but is prone to forgetting. The trade-off between stability and plasticity is a well-known constraint in CL (Mermillod et al., 2013). For CAL, we want the model to be plastic early on, and stable later on. We apply this intuition with $\lambda_t$: higher values indicate higher plasticity, since minimizing the classification error of samples from the current task is prioritized. Since $\mathcal{D}_{1:t-1}$ increases with $t$, $\lambda_t$ decreases and the model becomes more stable in later training rounds.

**Scaled Distillation w/ Submodular Sampling (CAL-SDS2)** CAL-SDS2 is another new CL approach we introduce in this work. CAL-SDS2 uses CAL-SD to regularize the model and uses a submodular sampling procedure to select a diverse representative set of history points to replay. Submodular functions are well-known to be suited to capture notions of diversity and representativeness (Lin & Bilmes, 2011; Wei et al., 2015; Bilmes, 2022) and the simple greedy algorithm can approximately maximize, under a cardinality constraint, a monotone submodular function up to a $1 - e^{-1}$ constant factor multiplicative guarantee (Fisher et al., 1978; Minoux, 1978; Mirzasoleiman et al., 2015). We define our submodular function $G$ as:

$$G(\mathcal{S}) \triangleq \sum_{x_i \in \mathcal{A}} \max_{x_j \in \mathcal{S}} w_{ij} + \lambda \log \left( 1 + \sum_{x_i \in \mathcal{S}} h(x_i) \right). \tag{3}$$

The first term is a facility location function, where $w_{ij}$ is a similarity score between samples $x_i$ and $x_j$. We use $w_{ij} = \exp\left(-\|z_i - z_j\|^2/2\sigma^2\right)$ where $z_i$ is the penultimate layer of model $f$ for $x_i$ and $\sigma$ is a hyperparameter.

The second term is a concave over modular function (Liu et al., 2013) and $h(x_i)$ is a standard AL measure of model uncertainty, such as entropy of the model's output distribution. Both terms are well known to be monotone non-decreasing submodular, as is their non-negatively weighted sum (Bilmes, 2022). The core reason for applying a concave function over a model-uncertainty-score-based modular function, instead of keeping it as a pure modular function, is to better align the modular uncertainty values with the facility location function. Otherwise, if we do not apply the concave function, the facility location function dominates during the early steps of the greedy algorithm and the modular function dominates in the later steps of greedy. In order to speed up SDS2 (to avoid the need to perform forward passes over the entire history before a single step), we randomly subsample the history indices to produce $\mathcal{A}$ and re-compute forward passes to produce fresh $z_i$ values for $i \in \mathcal{A}$ before re-constructing $G(\mathcal{S})$, and we then perform submodular maximization; thus $\mathcal{S} \subset \mathcal{A} \subset \mathcal{D}_{1:t-1}$. The objective of CAL-SDS2 is to ensure that the set of samples that are replayed are both difficult and diverse, similar to the motivation of the heuristic employed in Wei et al. (2015).

**Baselines** All proposed CAL methods are compared against two AL baselines. The first baseline is standard active learning, denoted as AL, which is no different from the procedure shown in Algorithm 1. We also consider active learning with warm starting (AL w/ WS), which uses the converged model from the previous round to initialize the model for the current round. Both models retrain on the entire dataset at each query round.

## 5   Experiments and Results

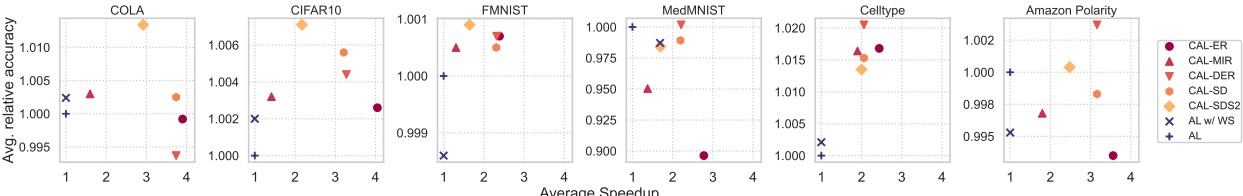

Figure 4: Relative accuracy vs. speedup averaged over different labeling budgets. For every dataset, we have *at least one* CAL method, that is faster with same accuracy than baseline (*top right region is desiderata*).

We evaluate the validation performance of the model when we train on different fractions ($b/n$) of the full dataset. We compute the *speedup* attained by a CAL method by dividing the wall-clock training time of the baseline AL method over the wall-clock training time of the CAL method. We test the CAL methods on a variety of different datasets spanning multiple modalities. Two baselines do not utilize CAL which are standard *AL* (active learning) as well as *AL w/ WS* (active learning with warm starting but still training using all the presently available labeled data).

Our objective is to demonstrate: (1) *at least one* CAL-based method exists that can match or outperform a standard active learning technique while achieving a significant speedup for every budget and dataset and (2) models that have been trained using a CAL-based method behave no differently than standard models. We emphasize that the purpose of this work is *not* to champion a single method, but rather to showcase an assortment of approaches in the CAL paradigm that achieve different performance/speedup trade-offs. Lastly, we would like to point out that some of the CAL methods are ablations of each other. For example, CAL-ER is ablation for CAL-DER (or CAL-SD) when we replace the distillation component. Similarly, CAL-SD is ablation of CAL-SDS2, where we remove the submodular selection part.

### 5.1   Datasets and Experimental Setup

We use the following datasets, which span a spectrum of data modalities, scale (both in terms of dataset size, and model's computational/memory footprint), and class balance.

**FMNIST:** The FMNIST dataset consists of 70,000 28×28 grayscale images of fashion items belonging to 10 classes (Xiao et al., 2017). A ResNet-18 architecture (He et al., 2016) with SGD is used. We apply data augmentation, as in Beck et al. (2021), consisting of random horizontal flips and random croppings.

**CIFAR-10:** CIFAR-10 consists of 60,000 32×32 color images with 10 different categories (Krizhevsky, 2009). We use a ResNet-18 and use the SGD optimizer for all CIFAR-10 experiments. We apply data augmentations consisting of random horizontal flips and random croppings.

**MedMNIST:** We use the DermaMNIST dataset within the MedMNIST collection (Yang et al., 2021a;b) for performance evaluation of CAL on medical imaging modalities. It consists of 3-color channel dermatoscopy images of 7 different skin diseases, originally obtained from Codella et al. (2019); Tschandl et al. (2018). A ResNet-18 architecture is used for all DermaMNIST experiments.

**Amazon Polarity Review:** (Zhang et al., 2015) is an NLP dataset consisting of reviews from Amazon and their corresponding star-ratings (5 classes) which was used for active learning in Coleman et al. (2020b). Similar to the previous work we consider a total unlabeled pool of size 2 million sentences and use a VDCNN-9 (Schwenk et al., 2017) architecture trained using Adam.

**COLA:** COLA (Warstadt et al., 2018) aims to check the linguistic acceptability of a sentence via binary classification. We consider an unlabeled size-7000 pool similar to Ein-Dor et al. (2020) and use a BERT (Devlin et al., 2019) backbone trained using Adam.

**Single-Cell Cell Type Identity Classification:** Recent single-cell RNA sequencing (scRNA-seq) technologies have enabled large-scale characterization of hundreds of thousands to millions of cells in complex tissues, and accurate cell type annotation is a crucial step in the study of such datasets. To this end, several deep learning models have been proposed to automatically label new scRNA-seq datasets (Xie et al., 2021). The HCL dataset is highly *class-imbalanced* and consists of scRNA-seq data for 562,977 cells across 63 cell types represented in 56 human tissues (Han et al., 2020). We use the ACTINN model (Ma & Pellegrini, 2019), a four-layer multi-layer perceptron that predicts the cell-type for each cell given its expression of 28832 genes, and uses an SGD optimizer.

**Hyperparameters:** Details about the specific hardware and the choices of hyperparameters used to train models for each technique can be found in Appendix A.5.

**Active Learning setup:** As done in previous work (Coleman et al., 2020b; Killamsetty et al., 2021a) for CIFAR10 and Amazon polarity review, budgets go from 10% to 50% in increments of 10% (results for other query sizes are presented in Appendix C.3). For FMNIST, MedMNIST, and Cell-type datasets, it goes from 10% to 30% in increments of 5%. Lastly, for COLA, we follow a budget using absolute sizes from 200 to 1000 in increments of 200 (similar to Ein-Dor et al. (2020)).[2] We adopt the AL framework proposed in Beck et al. (2021) for all experiments. In the main paper, we here present results for an uncertainty sampling-based acquisition function. *However, we provide results using other acquisition functions in Appendix C.*

## 5.2 Performance vs Speedup

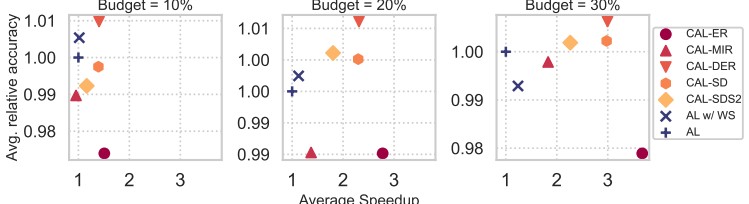

Figure 5: Relative accuracy vs. speedup averaged over different datasets. For every given budget, we have *at least one* CAL method that is faster with the same (if not better) accuracy than the baseline.

Our goal is to show that for every dataset and budget, there exists at least one CAL method that performs equivalent (if not better) than baseline AL. However, with raw accuracies, it is difficult to compare different CAL methods and baseline AL over different datasets at different budgets. Therefore, we begin by observing the relative gain in accuracy over the AL baseline. Relative gains further make it feasible to take an average across the budgets, for a given dataset, and to take averages across the datasets, for a given budget. For *every budget*, we normalize the accuracies of each method by that of baseline AL. This makes the baseline accuracy always **1**, irrespective of budget and dataset. Relative performances greater than 1 indicate better than baseline accuracy (and the opposite for the less than 1 case). Having said that: (1) keeping the budgets fixed to 10%, 20%, and 30% and averaging over the datasets (except COLA, since it has a different budget)

---

[2]We follow the query set sizes from (Ein-Dor et al., 2020)

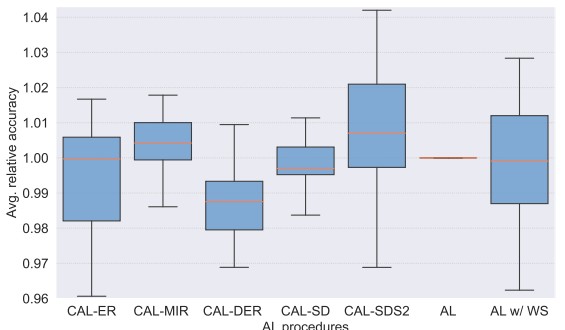

Figure 6: Comparison of CAL methods with the baseline on CIFAR-10C; absolute accuracies on individual benchmark and its average difference with the baseline AL are reported in Appendix A.2 table 8 and 10, respectively. All CAL methods perform within a standard deviation of the standard AL baseline, and CAL-SDS2 achieves the highest robust accuracy on average.

give us Figure 5; (2) keeping the dataset fixed, averaging the relative accuracy vs. speedups across different budgets give us Figure 4; and (3) further averaging the above across different datasets give us Figure 1. Methods in the top right corner are preferable. For space reasons, we display only relative accuracies in the main paper, but all absolute accuracies and standard deviations for each method for every dataset and every budget are available in Appendix A.2.

From the overall and per-dataset depiction of CAL's performance (Figures 1 and 4, respectively), it is evident that *there exists a CAL method that attains a significant speedup over a standard AL technique for every dataset and budget while preserving test set accuracy.* From Figure 4, we can further see that for some datasets (such as FMNIST and CIFAR-10), CAL-ER, a non-distillation, and uniform sampling-based method, only incur a minor drop in performance but attain the highest speedup. This suggests that naively biasing learning towards recent tasks can be sufficient to adapt the model to a new set of points between AL rounds. However, as we show in Figure 5 it is not universally true for all the datasets (at different budgets). Hence, the methods which include some type of distillation term (CAL-DER, CAL-SD, CAL-SDS2) generally perform the best out of all CAL methods. We believe that the submodular sampling-based method (CAL-SDS2) can be accelerated using stochastic methods and results improved by considering other submodular functions, which we leave as future work. It should be mentioned, however, that the concave function on $h(x_i)$ was essential for CAL-SDS2's performance.

## 5.3 Comparison Between Standard and CAL Models

In this section, we next assess whether CAL training has any adverse effect on the final model's behavior. We first demonstrate that CAL does *not* result in any deterioration of model robustness (Section 5.3.1). We then demonstrate that CAL models and baseline trained models are uncertain about a similar set of unseen examples (Section 5.3.2). Lastly, in appendix A.5 we provide a sensitivity analysis of our proposed methods, where we demonstrate that *CAL methods are robust to the changes to the hyperparameters.*

### 5.3.1 Robustness

Test time distributions can vary from training distributions, so it is important to ensure that models can generalize across different domains. Since models trained using CAL methods require significantly fewer gradient steps, the modified training procedure may produce fickler models that are less robust to domain shifts. To ensure against this, we evaluate CAL-method-trained model robustness in this section. We consider CIFAR-10C Hendrycks & Dietterich (2019), a dataset comprising 19 different corruptions each done at 5 levels of severity. For each model trained up to a 50% budget, we report the average classification accuracy over each corruption and compare it against the baseline in Figure 6; each result is an average of over three random seeds. We note that most of the CAL methods perform statistically similarly to standard active learning, all while providing significant acceleration. Moreover, on average across all the tests, **models trained with CAL-SDS2 are better than the models trained with CAL-DER**, where we see a difference of as much as 5% in corruptions such as *glass blur*; please refer to appendix A.3 table 8, table 9 and 10 for the absolute accuracies and statistical tests. Submodular sampling replays a diverse representative subset of history which is likely the reason behind CAL-SDS2's better robustness. The relationship of diversity with robustness has also been explored in previous works including Killamsetty et al. (2021b); Fang et al. (2022); Rozen et al. (2019); Gong et al. (2018).

### 5.3.2 Correlation of Uncertainty Scores

For models trained using CAL techniques to be used as valid substitutes for standard AL models, these two classes of models need to query similar samples at each AL round. This is particularly important if AL is being used solely as a data subset selection procedure (where the user is concerned about the quality of the resulting labeled dataset as opposed to the final model). When using uncertainty sampling as the AL acquisition function, computing the Pearson correlation between the entropy scores of baseline and CAL models on the validation set after every query round is one way of determining this. Since most AL policies incorporate some notion of uncertainty, we hypothesize that the results of this experiment should extend to other acquisition functions as well. A similar analysis is done in Coleman et al. (2020b).

As seen in Figure 7, the Pearson correlation between all pairs of models is positive at every AL query round. Thus, the nature of samples that the models are uncertain about, and thus are likely to be chosen at each round, is similar between the CAL-trained models and baseline AL-trained models. A breakdown of these correlations at every round is provided in Figure 14 in the Appendix.

## 6 Conclusion and Future Work

We proposed the CAL framework, the first method to circumvent the problem of having to retrain models between batch AL rounds. Across vision, natural language, medical imaging, and biological datasets, we show there is always a CAL-based method that either matches or outperforms standard AL while achieving considerable speedups. Since CAL is independent of the model architecture and AL strategy, this framework applies to a broad range of settings.

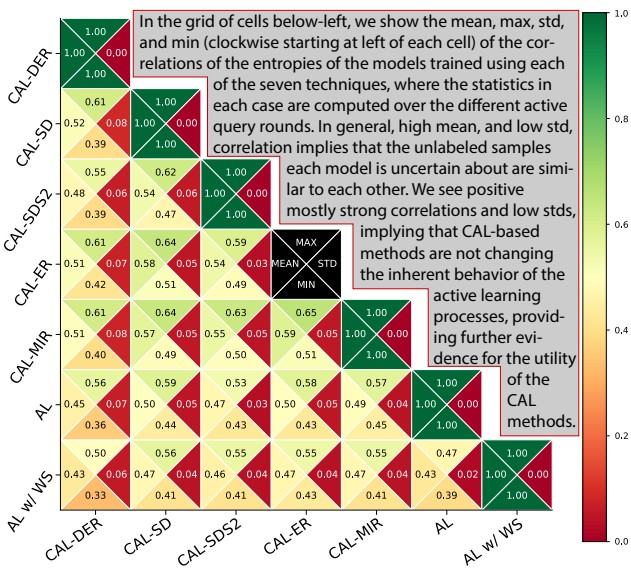

Figure 7: Cross-Method Entropy Correlation Statistics.

**Empirical Extensions** Future empirical directions may include the following: **(1):** CAL reduces the training time of the model, but not the AL query time although query time reductions could be an offshoot of this work; **(2):** CAL operates using existing AL query acquisition functions, but it is possible to tailor acquisition functions for CAL methods yielding further generalization and computational improvements; **(3):** while SDS2's use of submodularity helped robustness, additional submodular strategies can be used to further improve results and also for diverse AL query selection; and **(4):** CAL provides a novel application for CL; future CL work can be partially assessed based on its CAL performance.

**Theoretical Extensions** The discrepancy-based Active Learning framework is promising to establish generalization error bounds for the replay-based CAL framework. Specifically, leveraging the generalization results presented in Theorem 2 of Cui & Sato (2020), the upper bound on the risk function based on true data distribution (referred to as $\mathbb{P}$ in the theorem) is contingent upon the empirical distribution of the presently labeled data (referred to as $\mathbb{Q}$ in the theorem), that includes the newly acquired query batch.

In contrast to utilizing the entirety of the available labeled data, CAL strategically opts for a subset of this data through the employment of replay methods, for acceleration. Due to this, the empirical distribution of examples chosen by CAL (call it $\mathbb{Q}'$) could deviate from $\mathbb{Q}$. Consequently, one can modify the upper bound of the risk function based on true data distribution in Theorem 2 by adding the error term (which would depend on some notion of distance between $\mathbb{Q}$ and $\mathbb{Q}'$) which vanishes when we consider all the currently labeled examples, thus, recovering standard AL.

### Acknowledgments

This work was supported in part by the CONIX Research Center, one of six centers in JUMP, a Semiconductor Research Corporation (SRC) program sponsored by DARPA, by the National Science Foundation under

Grant Nos. IIS-2106937 and IIS-2148367, and by NIH/NHGRI U01 award HG009395. We thank Lilly Kumari, Tianyi Zhou, Shengjie Wang and all other MELODI lab members for their helpful discussions and feedback. We also thank the TMLR Action Editor and reviewers for their constructive comments.

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

# A  Additional Experimental Details on Main Results

## A.1  Procedure for normalized accuracy plots

Here we again explain the procedure to generate the normalized accuracy versus speedup plots, as reported in figures 1, 5, and 4. These plots help us to understand and compare the performance of different CAL methods against the baseline AL. For every dataset, we first get the respective accuracies and speedups of CAL methods and baseline AL, for every budget. This is also reported as a tabular form besides the scatter plot visualizations in section A.2.

For **every budget**, we normalize the accuracies of each method by that of baseline AL. This makes the baseline always at **1**, irrespective of budget and dataset. Relative performances greater than 1 indicate better than baseline accuracy (and similarly for the less than 1 case). Note that, for CIFAR10 and Amazon polarity review, budgets go from 10% to 50% in increments of 10%. However, for FMNIST, MedMNIST, and Celltype datasets, it goes from 10% to 30% in increments of 5%. Lastly for COLA, the budgets are different from the above, therefore, we don't include that when we average across the dataset at a fixed budget. Having said that,

- Keeping the budgets fixed to 10%, 20%, and 30% and averaging over the datasets (except COLA, since it has a different budget) will give us figure 5.

- Keeping the dataset fixed, averaging the relative accuracy v.s. speedups across different budgets will give us figure 4.

- Further averaging the above across different datasets will give us the main result figure 1.

From the results above, we can infer that CAL-DER and specialized methods such as CAL-SD/SDS2 always provide a speedup, all while preserving the accuracy, if not better.

## A.2  Results in Tabular Form

In this section, we expand our results mentioned in section 5.2. In particular, we report the absolute accuracies for each dataset, at every budget, and plot it against the observed speedup. All methods highlighted in blue are methods that use CAL. Note that all the results in this section are for uncertainty-based query pool acquisition functions. The choice of budget scale is taken from the previous works (Beck et al., 2021; Ein-Dor et al., 2020). Each dataset has a different complexity (and we train on them using different neural architectures), therefore they differ in the labeling budget.

**FMNIST**  Please refer to table 2 and figure 8.

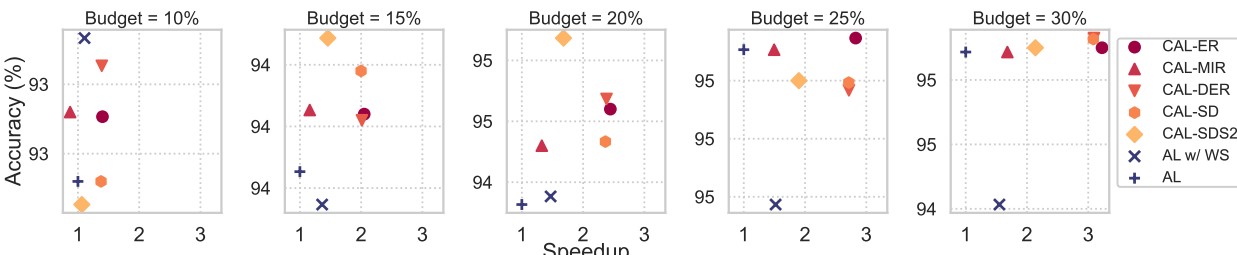

Figure 8: FMNIST Results

| Method | Test Accuracy (%) | | | | | Factor Speedup | | | | |
|---|---|---|---|---|---|---|---|---|---|---|
| | **10%** | **15%** | **20%** | **25%** | **30%** | **10%** | **15%** | **20%** | **25%** | **30%** |
| CAL-ER | 92.6 ± 0.1 | 93.9 ± 0.2 | 94.5 ± 0.1 | **94.9** ± 0.2 | **94.9** ± 0.2 | 1.5× | 1.4× | 2.0× | 2.4× | 2.8 × |
| CAL-MIR | 92.6 ± 0.3 | 93.9 ± 0.2 | 94.5 ± 0.0 | **94.9** ± 0.1 | **94.9** ± 0.0 | 0.9 × | 1.2× | 1.3× | 1.5× | 1.7× |
| CAL-DER | **92.7** ± 0.1 | 93.9 ± 0.1 | 94.5 ± 0.1 | 94.8 ± 0.2 | **94.9** ± 0.1 | 1.4 × | 2.0× | 2.4× | 2.7× | 3.1× |
| CAL-SD | 92.6 ± 0.1 | **94.0** ± 0.2 | 94.5 ± 0.1 | 94.8 ± 0.2 | **94.9** ± 0.1 | 1.4 × | 2.0× | 2.4× | 2.7× | 3.1× |
| CAL-SDS2 | 92.6 ± 0.1 | **94.0** ± 0.2 | **94.6** ± 0.2 | **94.9** ± 0.1 | **94.9** ± 0.1 | 1.1× | 1.5× | 1.7× | 1.9× | 2.1× |
| AL w/ WS | **92.7** ± 0.3 | 93.8 ± 0.2 | 94.4 ± 0.1 | 94.6 ± 0.1 | 94.4 ± 0.2 | 1.1× | 1.4× | 1.5× | 1.5× | 1.5× |
| AL | 92.6 ± 0.3 | 93.8 ± 0.0 | 94.4 ± 0.1 | **94.9** ± 0.2 | **94.9** ± 0.1 | 1.0× | 1.0× | 1.0× | 1.0× | 1.0× |

Table 2: FMNIST Results

**CIFAR10** Please refer to table 3 and figure 9.

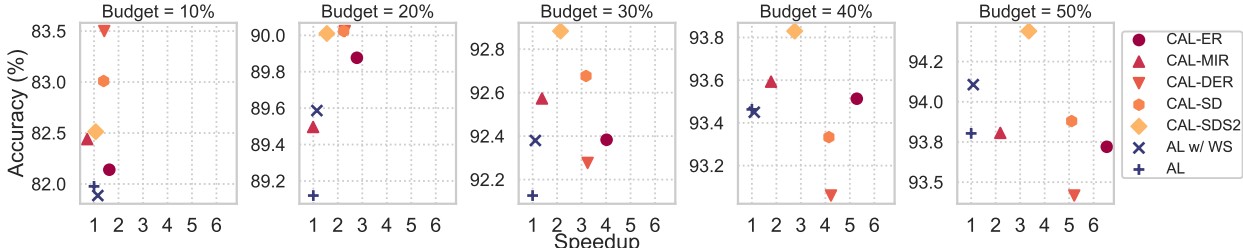

Figure 9: CIFAR-10 Results

| Method | Test Accuracy (%) | | | | | Factor Speedup | | | | |
|---|---|---|---|---|---|---|---|---|---|---|
| | **10%** | **20%** | **30%** | **40%** | **50%** | **10%** | **20%** | **30%** | **40%** | **50%** |
| CAL-ER | 82.1 ± 0.5 | 89.9 ± 0.3 | 92.4 ± 0.1 | 93.5 ± 0.1 | 93.7 ± 0.3 | 1.6× | 2.8× | 4.0× | 5.3× | 6.5× |
| CAL-MIR | 82.4 ± 0.4 | 89.5 ± 0.3 | 92.6 ± 0.3 | 93.6 ± 0.1 | 93.8 ± 0.2 | 0.7 × | 1.0× | 1.4× | 1.8× | 2.2× |
| CAL-DER | **83.5** ± 0.1 | 90.0 ± 0.4 | 92.3 ± 0.1 | 93.1 ± 0.2 | 93.4 ± 0.1 | 1.4× | 2.3× | 3.2× | 4.2× | 5.2× |
| CAL-SD | 83.0 ± 0.0 | 90.0 ± 0.4 | 92.7 ± 0.2 | 93.3 ± 0.3 | 93.9 ± 0.3 | 1.4× | 2.2× | 3.2× | 4.1× | 5.1× |
| CAL-SDS2 | 82.5 ± 0.1 | **90.1** ± 0.2 | **92.9** ± 0.4 | **94.0** ± 0.2 | **94.4** ± 0.1 | 1.1× | 1.6× | 2.1× | 2.7× | 3.4× |
| AL w/ WS | 81.9 ± 0.4 | 89.6 ± 0.5 | 92.4 ± 0.2 | 93.5 ± 0.1 | 94.1 ± 0.1 | 1.2× | 1.1× | 1.1× | 1.1× | 1.1× |
| AL | 82.0 ± 0.3 | 89.1 ± 0.2 | 92.1 ± 0.4 | 93.5 ± 0.3 | 93.8 ± 0.2 | 1.0× | 1.0× | 1.0× | 1.0× | 1.0× |

Table 3: CIFAR-10 Results

**MedMNIST** Please refer to table 4 and figure 10.

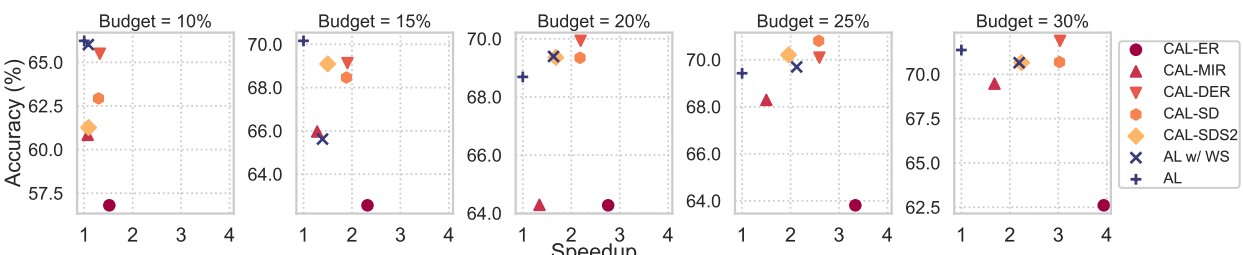

Figure 10: MedMNIST Results

| Method | Test Accuracy (%) | | | | | Factor Speedup | | | | |
|---|---|---|---|---|---|---|---|---|---|---|
| | **10%** | **15%** | **20%** | **25%** | **30%** | **10%** | **15%** | **20%** | **25%** | **30%** |
| CAL-ER | $64.7_{\pm 4.4}$ | $63.1_{\pm 5.2}$ | $68.4_{\pm 1.2}$ | $66.1_{\pm 3.7}$ | $67.5_{\pm 2.2}$ | $1.4\times$ | $2.0\times$ | $2.4\times$ | $2.9\times$ | $3.4\times$ |
| CAL-MIR | $64.1_{\pm 2.8}$ | $66.2_{\pm 1.0}$ | $69.2_{\pm 1.7}$ | $70.7_{\pm 1.1}$ | $71.8_{\pm 0.8}$ | $1.0\times$ | $1.2\times$ | $1.4\times$ | $1.6\times$ | $1.9\times$ |
| CAL-DER | $67.1_{\pm 2.9}$ | $67.9_{\pm 2.0}$ | $\mathbf{69.8}_{\pm 1.2}$ | $71.5_{\pm 0.6}$ | $72.2_{\pm 0.6}$ | $1.2\times$ | $1.6\times$ | $2.1\times$ | $2.6\times$ | $3.2\times$ |
| CAL-SD | $65.1_{\pm 3.1}$ | $67.8_{\pm 2.3}$ | $70.6_{\pm 0.9}$ | $71.1_{\pm 1.1}$ | $72.1_{\pm 1.1}$ | $1.2\times$ | $1.7\times$ | $2.0\times$ | $2.5\times$ | $2.9\times$ |
| CAL-SDS2 | $66.1_{\pm 3.6}$ | $68.1_{\pm 3.0}$ | $\mathbf{69.8}_{\pm 1.1}$ | $\mathbf{71.6}_{\pm 1.6}$ | $\mathbf{72.5}_{\pm 1.6}$ | $1.1\times$ | $1.5\times$ | $1.7\times$ | $2.0\times$ | $2.2\times$ |
| AL w/ WS | $67.0_{\pm 1.4}$ | $67.5_{\pm 0.7}$ | $69.5_{\pm 0.6}$ | $70.3_{\pm 1.0}$ | $71.3_{\pm 1.0}$ | $1.1\times$ | $1.3\times$ | $1.5\times$ | $1.7\times$ | $2.0\times$ |
| AL | $\mathbf{67.2}_{\pm 0.7}$ | $\mathbf{68.3}_{\pm 1.3}$ | $68.8_{\pm 0.8}$ | $69.0_{\pm 2.0}$ | $70.9_{\pm 1.5}$ | $1.0\times$ | $1.0\times$ | $1.0\times$ | $1.0\times$ | $1.0\times$ |

Table 4: MedMNIST Results

**Amazon Polarity Review**   Please refer to table 5 and figure 11.

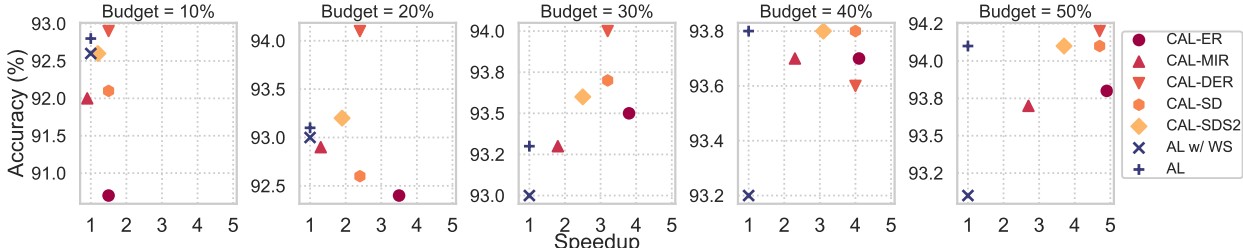

Figure 11: Amazon Polarity Results

| Method | Test Accuracy (%) | | | | | Factor Speedup | | | | |
|---|---|---|---|---|---|---|---|---|---|---|
| | **10%** | **20%** | **30%** | **40%** | **50%** | **10%** | **20%** | **30%** | **40%** | **50%** |
| CAL-ER | $90.7_{\pm 3.1}$ | $92.4_{\pm 1.2}$ | $93.5_{\pm 0.1}$ | $93.7_{\pm 0.2}$ | $93.8_{\pm 0.2}$ | 1.5x | 3.5x | 3.3x | 4.1x | 4.9x |
| CAL-MIR | $92.0_{\pm 0.9}$ | $92.9_{\pm 0.1}$ | $93.3_{\pm 0.3}$ | $93.7_{\pm 0.1}$ | $93.7_{\pm 0.2}$ | 0.9x | 1.3x | 1.8x | 2.3x | 2.7x |
| CAL-DER | $\mathbf{92.9}_{\pm 0.3}$ | $\mathbf{94.1}_{\pm 0.3}$ | $\mathbf{94.0}_{\pm 0.7}$ | $93.6_{\pm 0.8}$ | $\mathbf{94.2}_{\pm 0.3}$ | 1.5x | 2.4x | 3.2x | 4.0x | 4.7x |
| CAL-SD | $92.1_{\pm 0.3}$ | $92.6_{\pm 0.4}$ | $93.7_{\pm 0.1}$ | $\mathbf{93.8}_{\pm 0.1}$ | $94.1_{\pm 0.1}$ | 1.5x | 2.4x | 3.2x | 4.0x | 4.7x |
| CAL-SDS2 | $92.6_{\pm 0.3}$ | $93.2_{\pm 0.1}$ | $93.6_{\pm 0.1}$ | $\mathbf{93.8}_{\pm 0.4}$ | $94.1_{\pm 0.0}$ | 1.2x | 1.9x | 2.5x | 3.1x | 3.7x |
| AL w/ WS | $92.6_{\pm 0.5}$ | $93.0_{\pm 0.2}$ | $93.0_{\pm 0.1}$ | $93.2_{\pm 0.3}$ | $93.1_{\pm 0.1}$ | 1.0x | 1.0x | 1.0x | 1.0x | 1.0x |
| AL | $92.8_{\pm 0.2}$ | $93.1_{\pm 0.7}$ | $93.3_{\pm 1.1}$ | $\mathbf{93.8}_{\pm 0.5}$ | $94.1_{\pm 0.2}$ | 1.0x | 1.0x | 1.0x | 1.0x | 1.0x |

Table 5: Amazon Polarity Results

**COLA**   Please refer to table 6 and figure 12.

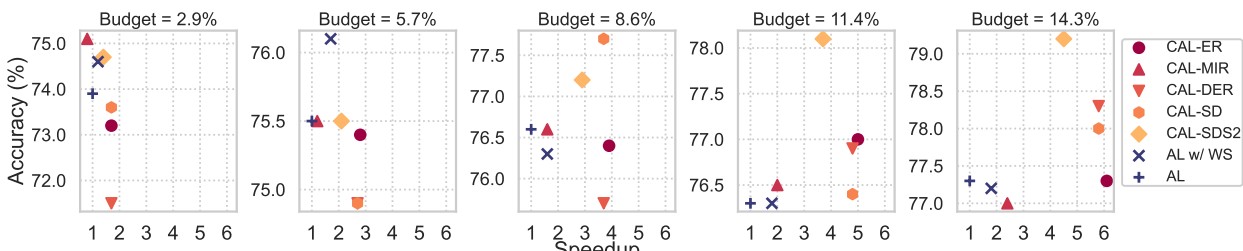

Figure 12: COLA Results

| Method | Test Accuracy (%) | | | | | Factor Speedup | | | | |
|---|---|---|---|---|---|---|---|---|---|---|
| | 2.9% | 5.7% | 8.6% | 11.4% | 14.3% | 2.9% | 5.7% | 8.6% | 11.4% | 14.3% |
| CAL-ER | $73.2_{\pm 1.7}$ | $75.4_{\pm 0.8}$ | $76.4_{\pm 1.0}$ | $77.0_{\pm 2.0}$ | $77.3_{\pm 1.4}$ | 1.7x | 2.8x | 3.9x | 5.0x | 6.1x |
| CAL-MIR | $\mathbf{75.1}_{\pm 0.2}$ | $75.5_{\pm 1.2}$ | $76.6_{\pm 1.0}$ | $76.5_{\pm 0.4}$ | $77.0_{\pm 0.3}$ | 0.8x | 1.2x | 1.6x | 2.0x | 2.4x |
| CAL-DER | $71.5_{\pm 2.7}$ | $74.9_{\pm 3.2}$ | $75.7_{\pm 1.5}$ | $76.9_{\pm 1.6}$ | $78.3_{\pm 0.8}$ | 1.7x | 2.7x | 3.7x | 4.8x | 5.8x |
| CAL-SD | $73.6_{\pm 1.9}$ | $74.9_{\pm 1.1}$ | $\mathbf{77.7}_{\pm 1.3}$ | $76.4_{\pm 0.3}$ | $78.0_{\pm 0.9}$ | 1.7x | 2.7x | 3.7x | 4.8x | 5.8x |
| CAL-SDS2 | $74.7_{\pm 2.8}$ | $75.5_{\pm 1.0}$ | $77.2_{\pm 0.9}$ | $\mathbf{78.1}_{\pm 0.8}$ | $\mathbf{79.2}_{\pm 0.5}$ | 1.4x | 2.1x | 2.9x | 3.7x | 4.5x |
| AL w/ WS | $74.6_{\pm 0.7}$ | $\mathbf{76.1}_{\pm 0.4}$ | $76.3_{\pm 1.0}$ | $76.3_{\pm 1.5}$ | $77.2_{\pm 0.9}$ | 1.2x | 1.7x | 1.6x | 1.8x | 1.8x |
| AL | $73.9_{\pm 2.9}$ | $75.5_{\pm 0.5}$ | $76.6_{\pm 2.0}$ | $76.3_{\pm 0.9}$ | $77.3_{\pm 1.6}$ | 1.0x | 1.0x | 1.0x | 1.0x | 1.0x |

Table 6: COLA Results.

**Single-Cell Cell-Type Identity** Please refer to table 7 and figure 13.

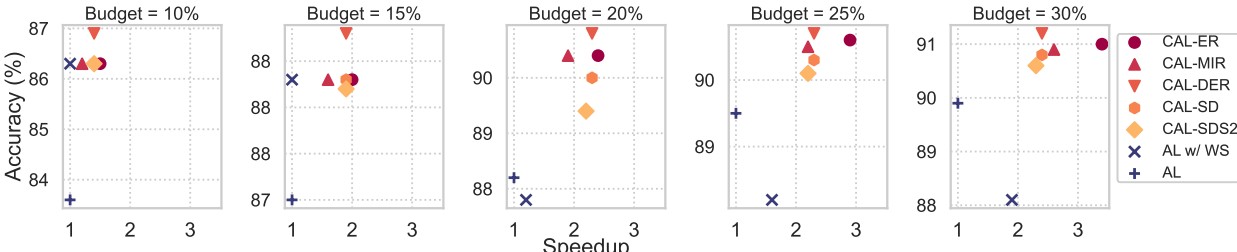

Figure 13: Single-Cell Cell-Type Identity Classification Results

| Method | Test Accuracy (%) | | | | | Factor Speedup | | | | |
|---|---|---|---|---|---|---|---|---|---|---|
| | 10% | 15% | 20% | 25% | 30% | 10% | 15% | 20% | 25% | 30% |
| CAL-ER | $86.3_{\pm 0.1}$ | $88.3_{\pm 0.1}$ | $89.7_{\pm 0.3}$ | $90.6_{\pm 0.2}$ | $91.0_{\pm 0.1}$ | $1.5\times$ | $2.0\times$ | $2.4\times$ | $2.9\times$ | $3.4\times$ |
| CAL-MIR | $86.3_{\pm 0.1}$ | $88.3_{\pm 0.1}$ | $89.7_{\pm 0.2}$ | $90.5_{\pm 0.2}$ | $90.9_{\pm 0.2}$ | $1.2\times$ | $1.6\times$ | $1.9\times$ | $2.2\times$ | $2.6\times$ |
| CAL-DER | $\mathbf{86.9}_{\pm 0.3}$ | $\mathbf{88.8}_{\pm 0.3}$ | $\mathbf{89.9}_{\pm 0.3}$ | $\mathbf{90.7}_{\pm 0.2}$ | $\mathbf{91.2}_{\pm 0.1}$ | $1.4\times$ | $1.9\times$ | $2.3\times$ | $2.8\times$ | $3.3\times$ |
| CAL-SD | $86.3_{\pm 0.1}$ | $88.3_{\pm 0.1}$ | $89.5_{\pm 0.2}$ | $90.3_{\pm 0.2}$ | $90.8_{\pm 0.2}$ | $1.4\times$ | $1.9\times$ | $2.3\times$ | $2.8\times$ | $3.3\times$ |
| CAL-SDS2 | $86.3_{\pm 0.1}$ | $88.2_{\pm 0.1}$ | $89.2_{\pm 0.3}$ | $90.1_{\pm 0.2}$ | $90.6_{\pm 0.1}$ | $1.4\times$ | $1.9\times$ | $2.3\times$ | $2.8\times$ | $3.3\times$ |
| AL w/ WS | $86.3_{\pm 0.1}$ | $88.3_{\pm 0.1}$ | $88.4_{\pm 0.8}$ | $88.2_{\pm 0.8}$ | $88.1_{\pm 0.8}$ | $1.0\times$ | $1.0\times$ | $1.2\times$ | $1.6\times$ | $1.9\times$ |
| AL | $83.6_{\pm 1.0}$ | $87.0_{\pm 0.3}$ | $88.6_{\pm 0.1}$ | $89.5_{\pm 0.2}$ | $89.9_{\pm 0.3}$ | $1.0\times$ | $1.0\times$ | $1.0\times$ | $1.0\times$ | $1.0\times$ |

Table 7: Single-Cell Cell-Type Identity Classification Results

### A.3 Out-of-distribution (OOD) generalization

Here we show results from the figure 6 in a tabular form in table 8. We compare the performance of CAL methods with baseline AL for OOD generalization. Reported mean values and standard deviations are computed over three different random seeds. As shown in the figure 6 in the main paper, and the table below, CAL methods are as robust to perturbations as baseline AL, despite the speedup. Finally, in table 10 we report that CAL-SDS2 is better compared to other CAL methods, on average.

| Corruption | CAL-ER | CAL-MIR | CAL-DER | CAL-SD | CAL-SDS2 | AL | AL w/ WS |
|---|---|---|---|---|---|---|---|
| saturate | $90.4_{\pm 0.4}$ | $90.7_{\pm 0.2}$ | $90.4_{\pm 0.1}$ | $90.5_{\pm 0.4}$ | $90.8_{\pm 0.2}$ | $90.7_{\pm 0.4}$ | $90.8_{\pm 0.2}$ |
| impulse noise | $52.4_{\pm 1.4}$ | $53.4_{\pm 0.8}$ | $53.3_{\pm 3.9}$ | $53.6_{\pm 2.6}$ | $55.8_{\pm 1.3}$ | $54.5_{\pm 2.9}$ | $48.7_{\pm 2.7}$ |
| defocus blur | $79.6_{\pm 1.3}$ | $81.6_{\pm 0.4}$ | $79.3_{\pm 1.2}$ | $80.6_{\pm 1.1}$ | $79.6_{\pm 1.7}$ | $80.9_{\pm 1.0}$ | $80.4_{\pm 0.7}$ |
| contrast | $74.6_{\pm 0.3}$ | $76.1_{\pm 0.8}$ | $73.8_{\pm 0.9}$ | $75.1_{\pm 1.0}$ | $74.2_{\pm 0.6}$ | $77.1_{\pm 1.4}$ | $76.4_{\pm 2.5}$ |
| frost | $76.0_{\pm 0.4}$ | $76.1_{\pm 2.0}$ | $73.8_{\pm 1.0}$ | $75.4_{\pm 1.5}$ | $76.9_{\pm 1.2}$ | $75.4_{\pm 1.6}$ | $77.0_{\pm 0.3}$ |
| speckle noise | $61.7_{\pm 1.7}$ | $61.7_{\pm 2.6}$ | $60.9_{\pm 3.7}$ | $61.6_{\pm 3.8}$ | $64.3_{\pm 0.5}$ | $61.7_{\pm 0.3}$ | $59.4_{\pm 1.0}$ |
| pixelate | $74.4_{\pm 1.3}$ | $73.9_{\pm 2.2}$ | $75.2_{\pm 0.9}$ | $74.8_{\pm 1.6}$ | $76.5_{\pm 0.7}$ | $75.9_{\pm 1.3}$ | $77.0_{\pm 1.3}$ |
| zoom blur | $74.4_{\pm 1.0}$ | $76.7_{\pm 1.0}$ | $74.4_{\pm 2.3}$ | $75.4_{\pm 1.5}$ | $74.1_{\pm 2.8}$ | $75.9_{\pm 1.9}$ | $75.3_{\pm 1.0}$ |
| elastic transform | $82.5_{\pm 0.1}$ | $83.5_{\pm 0.2}$ | $82.0_{\pm 0.5}$ | $82.1_{\pm 0.8}$ | $82.6_{\pm 0.6}$ | $82.0_{\pm 1.3}$ | $82.9_{\pm 0.2}$ |
| spatter | $81.8_{\pm 0.7}$ | $82.9_{\pm 1.2}$ | $82.4_{\pm 0.6}$ | $82.6_{\pm 0.5}$ | $82.8_{\pm 0.8}$ | $83.0_{\pm 1.0}$ | $82.5_{\pm 1.2}$ |
| snow | $80.3_{\pm 0.4}$ | $80.2_{\pm 0.7}$ | $79.5_{\pm 0.8}$ | $79.7_{\pm 0.7}$ | $80.8_{\pm 0.1}$ | $80.1_{\pm 0.8}$ | $80.6_{\pm 0.3}$ |
| fog | $86.8_{\pm 0.3}$ | $86.9_{\pm 0.3}$ | $85.9_{\pm 0.9}$ | $86.3_{\pm 0.3}$ | $86.5_{\pm 0.4}$ | $86.6_{\pm 0.3}$ | $87.9_{\pm 0.7}$ |
| Gaussian noise | $46.6_{\pm 1.0}$ | $46.6_{\pm 3.3}$ | $44.9_{\pm 6.3}$ | $46.7_{\pm 4.7}$ | $50.7_{\pm 0.9}$ | $45.9_{\pm 0.4}$ | $43.2_{\pm 1.1}$ |
| brightness | $92.1_{\pm 0.3}$ | $92.4_{\pm 0.3}$ | $92.1_{\pm 0.3}$ | $92.2_{\pm 0.1}$ | $92.5_{\pm 0.1}$ | $92.7_{\pm 0.1}$ | $92.6_{\pm 0.2}$ |
| Gaussian blur | $69.7_{\pm 2.2}$ | $72.3_{\pm 1.3}$ | $69.3_{\pm 2.5}$ | $71.3_{\pm 1.5}$ | $69.3_{\pm 2.8}$ | $71.6_{\pm 1.6}$ | $70.4_{\pm 1.3}$ |
| motion blur | $75.1_{\pm 1.2}$ | $77.1_{\pm 1.2}$ | $74.4_{\pm 0.9}$ | $75.1_{\pm 1.6}$ | $74.5_{\pm 0.7}$ | $74.7_{\pm 0.6}$ | $76.8_{\pm 0.4}$ |
| shot noise | $58.6_{\pm 1.5}$ | $58.5_{\pm 2.9}$ | $57.3_{\pm 4.8}$ | $58.6_{\pm 4.1}$ | $61.6_{\pm 0.5}$ | $58.3_{\pm 0.2}$ | $56.1_{\pm 1.0}$ |
| jpeg compression | $79.0_{\pm 1.5}$ | $78.4_{\pm 0.5}$ | $78.5_{\pm 0.2}$ | $78.6_{\pm 0.5}$ | $79.1_{\pm 0.1}$ | $77.7_{\pm 0.1}$ | $77.7_{\pm 0.1}$ |
| glass blur | $51.8_{\pm 1.4}$ | $54.6_{\pm 3.0}$ | $48.6_{\pm 2.8}$ | $50.7_{\pm 2.1}$ | $53.9_{\pm 2.4}$ | $49.4_{\pm 3.1}$ | $52.6_{\pm 2.1}$ |

Table 8: Accuracy (in %) comparison of CAL methods with the baseline on the CIFAR-10C dataset. Results were reported as an average over three random seeds. Models trained with CAL procedure perform statistically similar to the one trained with baseline AL.

| Corruption | CAL-ER/AL | CAL-MIR/AL | CAL-DER/AL | CAL-SD/AL | CAL-SDS2/AL |
|---|---|---|---|---|---|
| saturate | 0.3957 | 0.9104 | 0.2118 | 0.4516 | 0.7961 |
| impulse noise | 0.3209 | 0.5692 | 0.7029 | 0.7134 | 0.5374 |
| defocus blur | 0.2477 | 0.2794 | 0.1561 | 0.7854 | 0.3178 |
| contrast | **0.0335 (+)** | 0.3023 | **0.0256 (-)** | 0.1020 | **0.0250 (-)** |
| frost | 0.5793 | 0.6639 | 0.2147 | 0.9548 | 0.2778 |
| speckle noise | 0.9873 | 0.9922 | 0.7409 | 0.9739 | **0.0017 (+)** |
| pixelate | 0.2243 | 0.2521 | 0.4661 | 0.3913 | 0.5636 |
| zoom blur | 0.2824 | 0.5567 | 0.4279 | 0.7238 | 0.3910 |
| elastic transform | 0.5482 | 0.1295 | 0.9975 | 0.9154 | 0.5176 |
| spatter | 0.1504 | 0.9401 | 0.4310 | 0.5843 | 0.7800 |
| snow | 0.7823 | 0.8905 | 0.3747 | 0.5455 | 0.1976 |
| fog | 0.4715 | 0.2534 | 0.2781 | 0.4512 | 0.7142 |
| gaussian noise | 0.3514 | 0.7485 | 0.7836 | 0.8056 | **0.0013 (+)** |
| brightness | **0.0309 (-)** | 0.2042 | **0.0341 (-)** | **0.0091 (-)** | **0.0474 (-)** |
| gaussian blur | 0.2969 | 0.5867 | 0.2605 | 0.8658 | 0.2997 |
| motion blur | 0.6474 | **0.0368 (+)** | 0.6263 | 0.6801 | 0.7135 |
| shot noise | 0.7252 | 0.8920 | 0.7311 | 0.9091 | **0.0004 (+)** |
| jpeg compression | **0.0110 (+)** | 0.1080 | **0.0016 (+)** | **0.0286 (+)** | **0.0000 (+)** |
| glass blur | 0.2820 | 0.1047 | 0.7633 | 0.5807 | 0.1182 |

Table 9: Pairwise p-values were computed to compare each Confidence-Aware Learning (CAL) method with the standard Active Learning (AL) approach across various corruption tests. p-values less than 0.05 are highlighted in **bold**, and instances, where CAL methods outperform (+) or underperform (-) compared to standard AL, are indicated. Overall, our findings suggest that the disparities between CAL and AL methods are typically not statistically significant, implying that CAL generally does not compromise robustness. When a statistically significant difference does arise, CAL (particularly CAL-SDS2) tends to outperform AL more frequently than it falls short.

## A.4 Correlation between CAL and baseline AL

In figure 14, we provide the Pearson correlation between the uncertainty scores of models on the held-out test set, before every query round. A positive correlation between CAL models and baseline AL models

| Method | Average Accuracy difference (in %) |
|--------|:----------------------------------:|
| CAL-ER | -0.34 |
| CAL-MIR | 0.49 |
| CAL-DER | -0.96 |
| CAL-SD | -0.17 |
| CAL-SDS2 | 0.63 |
| AL w/ WS | -0.32 |

Table 10: The average difference in the accuracy (in %) of models trained with baseline AL from models trained with CAL, on different benchmarks across the CIFAR10-C dataset. Higher numbers are better. We can see that CAL-SDS2 is better compared to other CAL methods, on average. We hypothesize that this can be attributed to submodular sampling from history.

before every query round suggests that the nature of examples chosen by the CAL models is similar to that of baseline AL models. Note that each entry in the correlation matrix is averaged over three random seeds corresponding to the random initialization of each model.

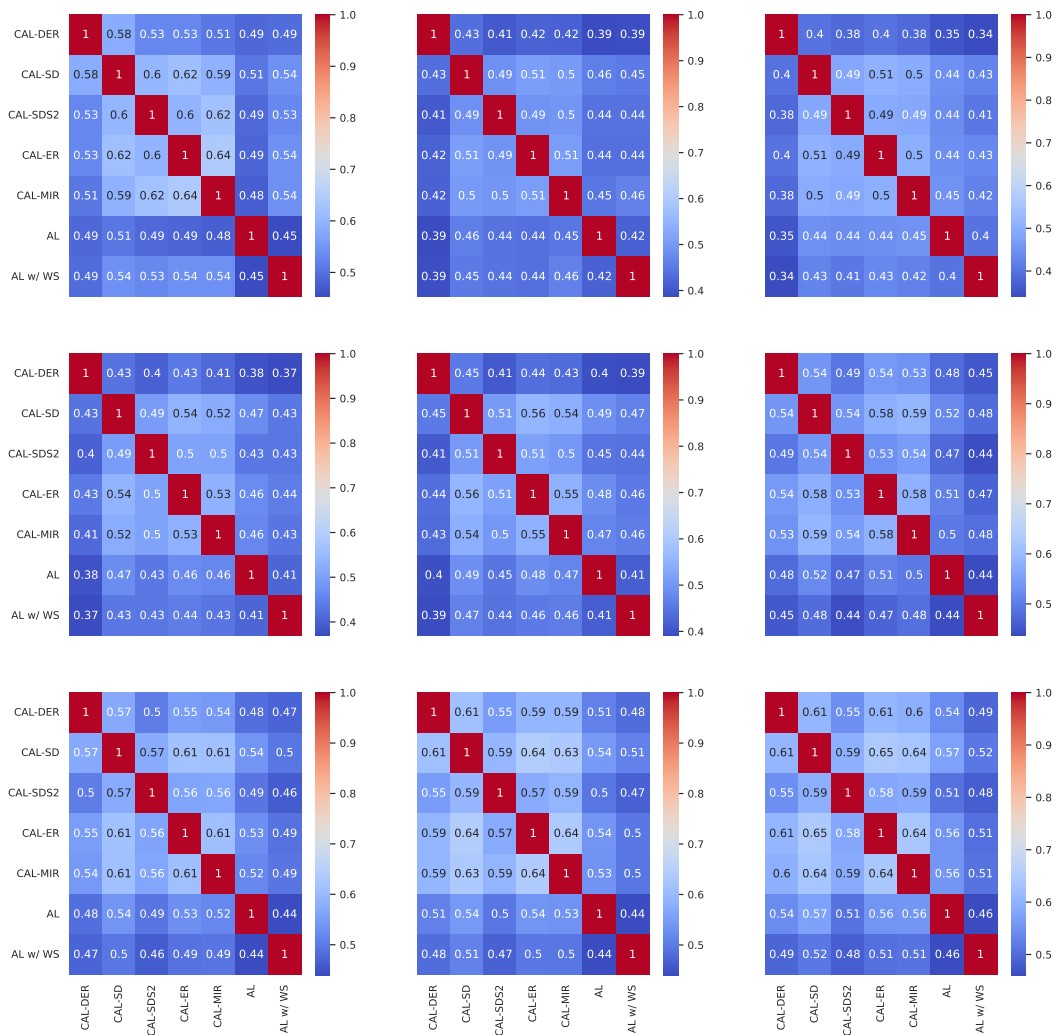

Figure 14: Pearson correlation between uncertainty scores of models on held out set after every query round. We are showing nine query rounds in row-major order (so the top left is after the first query round, the top middle is after the second, and so on). Positive correlation of uncertainty scores suggests that the nature of examples the models are uncertain about, and thus likely to be chosen at every query round, is similar between the CAL-trained models and baseline AL-trained models.

## A.5 Hyperparameters

For every dataset and every CAL/AL strategy, learning rate ($lr$) and batch size ($m$) are chosen based on whichever setting achieves the highest performance on standard AL. This means CAL methods can still improve if we tune either of learning rate or batch size. Replay size is critical for the performance of continual learning algorithms. On the one hand, we do not want the replay batch size, $m^{(h)}$, to be too small since then we will forget some history. But on the other hand, we also do not want $m^{(h)}$ to be too large since there will be a computational cost associated with that. Considering the mentioned constraints, for all CAL methods, we, therefore, set replay size as $m^{(h)} \in \{m, 2m\}$ (used in all CAL methods). Via experimentation on a subset of the datasets, we found $m^{(h)}$ less than or greater than this range suffered either from forgetting (when too small) or extra computation without an accuracy benefit (when too large). We set $\alpha \in \{0.1, 0.25, 0.5, 0.75\}$ (used in CAL-DER, CAL-SD, and CAL-SDS2), $\beta \in \{0.75, 1\}$ (used in CAL-DER). The scale and size of the search space for $\alpha$ and $\beta$ is inspired from Buzzega et al. (2020). Lastly, $\sigma \in \{0.1, 1\}$ (used in CAL-SDS2), and $\lambda \in \{0.1, 1, 10\}$ (used in CAL-SDS2).

We select the configuration for each CAL model that achieves the highest accuracy. "$c$" is the hyperparameter used in CAL-MIR and CAL-SDS2 to subsample the history before finding the $m^{(h)}$ samples to replay, but this parameter is not tuned for any of the presented results. We list the specific set of hyperparameters we use for all the main experimental results in this section.

### A.5.1   FMNIST

All experiments for FMNIST used a ResNet-18 with an SGD optimizer, with learning rate of 0.01 and batch size of 64. For all the CAL methods, we fix $m^{(h)} = 128$. A NVIDIA GeForce RTX 1080 GPU was used to run all the reported experiments.

**CAL-MIR**   $c = 256$

**CAL-DER**   $\alpha = 0.1$, $\beta = 1$

**CAL-SD**   $\alpha = 0.25$

**CAL-SDS2**   $c = 256$, $\alpha = 0.25$, $\sigma = 0.1$, $\lambda = 1$

### A.5.2   CIFAR-10

All experiments for CIFAR-10 used a ResNet-18 with an SGD optimizer, with learning rate of 0.02 and a batch size of 20. For all the CAL methods, we fix $m^{(h)} = 40$. Training is done on an NVIDIA GeForce RTX 2080.

**CAL-MIR**   $c = 100$

**CAL-DER**   $\alpha = 0.1$, $\beta = 1$

**CAL-SD**   $\alpha = 0.25$

**CAL-SDS2**   $c = 100$, $\alpha = 0.25$, $\sigma = 0.1$, $\lambda = 0.1$

### A.5.3   MedMNIST

All experiments for MedMNIST used a ResNet-18 with an Adam optimizer, with learning rate of 0.001 and a batch size of 128. For all CAL methods, we fix $m^{(h)} = 128$. All reported models were trained on an NVIDIA GeForce RTX 2080.

**CAL-MIR**   $c = 270$

**CAL-DER**   $\alpha = 0.1$, $\beta = 1$

**CAL-SD**   $\alpha = 0.5$

**CAL-SDS2**   $c = 270$, $\alpha = 0.5$, $\sigma = 0.1$, $\lambda = 10$

### A.5.4   Amazon Polarity Review

Throughout our experiments, we sample 2M sentences and use them as the total training set instead. We use Adam optimizer with default parameters with a learning rate of 0.001 and a batch size of 128 for 6 epochs. For all the CAL methods, we fix $m^{(h)} = 128$. All reported models were trained on an NVIDIA GeForce 1080 Ti.

**CAL-MIR**   $c = 256$,

**CAL-DER**   $\alpha = 0.25$, $\beta = 0.75$

**CAL-SD**   $\alpha = 0.5$

**CAL-SDS2**   $c = 256$, $\alpha = 0.75$, $\sigma = 1$, $\lambda = 1$

### A.5.5   COLA

For all of our experiments we use Huggingface's transformer library Wolf et al. (2020) and use a maximum sentence length of 100. We use Adam optimizer and a learning rate of $5 \cdot 10^{-5}$, use a batch size of 25 and $m^{(h)} = 25$. Models were trained on a single NVIDIA GeForce 1080 Ti.

**CAL-MIR**   $c = 50$

**CAL-DER**   $\alpha = 0.25$, $\beta = 0.75$

**CAL-SD**   $\alpha = 0.75$

**CAL-SDS2**   $c = 50$, $\alpha = 0.5$, $\sigma = 1$, $\lambda = 1$.

### A.5.6   Single-Cell Cell-Type Identity Classification

All experiments use SGD optimizer with standard parameters with learning rate of 0.001 and a batch size 128. For all the CAL methods, we fix $m^{(h)} = 128$. Training is done on an NVIDIA A100-PCIE-40GB.

**CAL-MIR**   $c = 200$,

**CAL-DER**   $\alpha = 0.1$, $\beta = 1$

**CAL-SD**   $\alpha = 1$

**CAL-SDS2**   $c = 100$, $\alpha = 0.25$, $\sigma = 0.1$, $\lambda = 1$

### A.6   Hyperparameter Sensitivity Analysis for Proposed Methods

In this section, we provide a hyperparameter sensitivity analysis methods for CAL-SD and CAL-SDS2 on CIFAR-10. For CAL-SD, only $\alpha$ is tuned and the best value is used for CAL-SDS2. Therefore, we tuned 1 hyperparameter for CAL-SD while $\sigma$ and $\lambda$ are tuned for CAL-SDS2. Note that the differences in final test accuracy at 50% budget on CIFAR-10 across different configurations are negligible, as shown in the tables below.

| $\alpha$ | 0.1 | 0.25 | 0.75 | 0.9 |
|---|---|---|---|---|
| | 93.79 | 93.90 | 93.58 | 93.38 |

Table 11: CAL-SD Sensitivity

| $\sigma$ \ $\lambda$ | 0.1 | 0.5 | 1 | 5 | 10 |
|---|---|---|---|---|---|
| 0.1 | 94.20 | 94.27 | 94.32 | 94.34 | 94.16 |
| 1 | 94.33 | 94.22 | 94.28 | 94.15 | **94.44** |
| 10 | 94.32 | 94.29 | 94.27 | 94.30 | 94.18 |

Table 12: CAL-SDS2 Sensitivity

# B    A primer on Continual Learning

We define $\mathcal{D}_{1:n} = \bigcup_{i \in [n]} \mathcal{D}_i$. In CL, the dataset consists of $T$ tasks $\{\mathcal{D}_1, ..., \mathcal{D}_T\}$ that are presented to the model sequentially, where $\mathcal{D}_t = \{(x_i, y_i)\}_{i \in N_t}$, $N_t$ are the task-$t$ sample indices, and $n_t = |N_t|$. At time $t \in [T]$, the data/label pairs are sampled from the current task $(x, y) \sim \mathcal{D}_t$, and the model has only limited access to the history $\mathcal{D}_{1:t-1}$. If the model is trained only on $\mathcal{D}_t$ using standard optimization algorithms, the model will exhibit catastrophic forgetting. The CL objective is to efficiently adapt the model to $\mathcal{D}_t$ while retaining the performance on the history. Given a loss function $\ell : \mathcal{X} \times \mathcal{Y} \mapsto \mathbb{R}$, initial parameters $\theta_{t-1}$, and a model $f$, $\theta_t$ can be obtained as the solution to the CL optimization problem (Aljundi et al., 2019b; Chaudhry et al., 2019; Lopez-Paz & Ranzato, 2017):

$$\min_{\theta} \mathop{\mathbb{E}}_{(x,y) \sim \mathcal{D}_t} \ell(y, f(x;\theta))$$

$$\text{s.t.} \mathop{\mathbb{E}}_{(x',y') \sim \mathcal{D}_{1:t-1}} \ell\left(y', f(x';\theta)\right) \leq \mathop{\mathbb{E}}_{(x',y') \sim \mathcal{D}_{1:t-1}} \ell\left(y', f(x';\theta_{t-1})\right)$$

# C    Results for Additional Active Learning Strategies

In this section, we demonstrate that CAL methods can accelerate AL strategies other than entropy sampling without incurring any significant performance drops. We test multiple AL strategies on FMNIST Xiao et al. (2017) and CIFAR-10 Krizhevsky (2009). Note that the speedups are approximately the same as the ones reported in Section A since the training time is generally independent of the selected AL strategy.

## C.1    Overview of Strategies

**Margin Score Sampling**    This strategy is another form of uncertainty sampling Settles (2009) as described in the main paper. Instead of the entropy of $f(x;\theta)$, the margin score is used as the entropy score i.e., $h(x) \triangleq 1 - (f(x;\theta)_i - f(x;\theta)_j)$ where $i$ and $j$ are the indices corresponding to the highest and second highest values of $f(x;\theta)$ respectively.

**FASS**    FASS Wei et al. (2015) is a two-staged selection method that uses both uncertainty sampling and submodular maximization. Initially, a set of samples $\mathcal{A}$ of cardinality $c * b_t$ is chosen from $\mathcal{U}$ using uncertainty sampling, where $c > 1$ is a tuneable hyperparameter. Next, $U_t$ is constructed by greedily selecting samples that maximize a submodular set function $G : 2^{\mathcal{A}} \to \mathbb{R}_+$ defined on a ground set $\mathcal{A}$. Entropy is once again used as the uncertainty metric for the initial stage. For the second stage, $G$ is defined to be the facility location function Wei et al. (2015) expressed below:

$$G(\mathcal{S}) = \sum_{x_i \in \mathcal{A}} \max_{x_j \in \mathcal{S}} w_{ij}, \tag{4}$$

where $\mathcal{S} \subseteq \mathcal{A}$ and $w_{ij}$ is a similarity score between samples $x_i$ and $x_j$. In our experiments, $w_{ij} = \exp\left(-\|z_i - z_j\|^2 / 2\sigma^2\right)$ where $z_i$ is the penultimate layer representation of model $f$ for $x_i$ and $\sigma$ is a hyperparameter.

**GLISTER**    GLISTER Killamsetty et al. (2021a) solves a bi-level optimization problem in order to select samples to label. Specifically, GLISTER solves

$$\operatorname*{argmax}_{\mathcal{S} \subseteq \mathcal{U}_t, |\mathcal{S}| \leq b_t} LL_V(\operatorname*{argmax}_{\theta} LL_T(\theta, \mathcal{S}), \mathcal{V}) \tag{5}$$

where $LL_V$ is the log-likelihood on the validation set $\mathcal{V}$, and $LL_T$ is the log-likelihood on the subset $\mathcal{S}$.

### C.2 Results

| Method | Test Accuracy (%) | | | | |
|---|---|---|---|---|---|
| | **10%** | **15%** | **20%** | **25%** | **30%** |
| CAL-ER | **92.8** $\pm$ 0.1 | **94.1** $\pm$ 0.1 | 94.8 $\pm$ 0.1 | **95.1** $\pm$ 0.3 | **95.2** $\pm$ 0.2 |
| CAL-MIR | 92.6 $\pm$ 0.2 | **94.1** $\pm$ 0.4 | **94.9** $\pm$ 0.2 | 95.0 $\pm$ 0.2 | **95.2** $\pm$ 0.2 |
| CAL-DER | 91.8 $\pm$ 0.5 | 93.1 $\pm$ 0.1 | 94.3 $\pm$ 0.3 | 94.6 $\pm$ 0.1 | 94.8 $\pm$ 0.2 |
| CAL-SD | 92.5 $\pm$ 0.1 | 93.8 $\pm$ 0.1 | 94.8 $\pm$ 0.0 | **95.1** $\pm$ 0.2 | **95.2** $\pm$ 0.0 |
| CAL-SDS2 | 87.8 $\pm$ 1.1 | 93.4 $\pm$ 0.1 | 94.6 $\pm$ 0.1 | 95.0 $\pm$ 0.2 | **95.2** $\pm$ 0.1 |
| AL w/ WS | **92.8** $\pm$ 0.0 | 94.0 $\pm$ 0.3 | 94.6 $\pm$ 0.1 | 94.8 $\pm$ 0.1 | 95.0 $\pm$ 0.2 |
| AL | 92.7 $\pm$ 0.1 | **94.1** $\pm$ 0.3 | **94.9** $\pm$ 0.1 | 95.0 $\pm$ 0.2 | **95.2** $\pm$ 0.1 |

Table 13: FMNIST with Margin Score Sampling

| Method | Test Accuracy (%) | | | | |
|---|---|---|---|---|---|
| | **10%** | **20%** | **30%** | **40%** | **50%** |
| CAL-ER | 81.5 $\pm$ 0.1 | 89.3 $\pm$ 0.1 | 92.2 $\pm$ 0.2 | 93.4 $\pm$ 0.1 | 93.8 $\pm$ 0.0 |
| CAL-MIR | 81.9 $\pm$ 0.1 | 89.6 $\pm$ 0.2 | 92.2 $\pm$ 0.4 | 93.6 $\pm$ 0.0 | 94.0 $\pm$ 0.2 |
| CAL-DER | 83.0 $\pm$ 0.2 | 89.5 $\pm$ 0.2 | 92.2 $\pm$ 0.2 | 93.2 $\pm$ 0.2 | 93.6 $\pm$ 0.0 |
| CAL-SD | 82.6 $\pm$ 0.4 | 89.9 $\pm$ 0.4 | 92.4 $\pm$ 0.2 | 93.5 $\pm$ 0.1 | 93.8 $\pm$ 0.2 |
| CAL-SDS2 | 82.5 $\pm$ 0.2 | 90.2 $\pm$ 0.2 | 92.5 $\pm$ 0.2 | **93.8** $\pm$ 0.2 | **94.1** $\pm$ 0.1 |
| AL w/ WS | **83.1** $\pm$ 0.1 | **90.3** $\pm$ 0.3 | **93.0** $\pm$ 0.2 | 93.5 $\pm$ 0.3 | 93.6 $\pm$ 0.2 |
| AL | 75.1 $\pm$ 1.2 | 87.1 $\pm$ 1.0 | 90.2 $\pm$ 0.5 | 92.0 $\pm$ 0.0 | 92.8 $\pm$ 0.5 |

Table 14: CIFAR-10 with Margin Score Sampling

| Method | Test Accuracy (%) | | | | |
|---|---|---|---|---|---|
| | **10%** | **15%** | **20%** | **25%** | **30%** |
| CAL-ER | 92.6 $\pm$ 0.1 | **93.9** $\pm$ 0.2 | 94.6 $\pm$ 0.2 | **95.0** $\pm$ 0.1 | 94.9 $\pm$ 0.0 |
| CAL-MIR | 92.5 $\pm$ 0.1 | 93.8 $\pm$ 0.3 | 94.6 $\pm$ 0.1 | 94.8 $\pm$ 0.1 | 94.9 $\pm$ 0.2 |
| CAL-DER | 92.7 $\pm$ 0.1 | 93.8 $\pm$ 0.1 | 94.5 $\pm$ 0.1 | 94.7 $\pm$ 0.1 | **95.0** $\pm$ 0.2 |
| CAL-SD | **92.8** $\pm$ 0.1 | **93.9** $\pm$ 0.1 | **94.7** $\pm$ 0.1 | 94.8 $\pm$ 0.3 | 94.9 $\pm$ 0.1 |
| CAL-SDS2 | **92.8** $\pm$ 0.0 | 93.8 $\pm$ 0.2 | 94.5 $\pm$ 0.1 | 94.8 $\pm$ 0.2 | 94.9 $\pm$ 0.1 |
| AL w/ WS | 92.5 $\pm$ 0.1 | 93.8 $\pm$ 0.3 | 94.0 $\pm$ 0.2 | 94.3 $\pm$ 0.2 | 94.3 $\pm$ 0.0 |
| AL | 92.7 $\pm$ 0.4 | **93.9** $\pm$ 0.1 | 94.5 $\pm$ 0.1 | 94.7 $\pm$ 0.3 | 94.8 $\pm$ 0.1 |

Table 15: FMNIST with FASS

|  | Test Accuracy (%) | | | | |
|---|---|---|---|---|---|
| Method | 10% | 20% | 30% | 40% | 50% |
| CAL-ER | $82.2_{\pm 0.2}$ | $89.8_{\pm 0.2}$ | $92.5_{\pm 0.2}$ | $93.4_{\pm 0.4}$ | $93.7_{\pm 0.2}$ |
| CAL-MIR | $82.2_{\pm 0.3}$ | $89.4_{\pm 0.2}$ | $92.3_{\pm 0.1}$ | $93.4_{\pm 0.0}$ | $93.5_{\pm 0.1}$ |
| CAL-DER | $\mathbf{83.1}_{\pm 0.3}$ | $89.7_{\pm 0.2}$ | $91.9_{\pm 0.1}$ | $93.1_{\pm 0.2}$ | $93.5_{\pm 0.1}$ |
| CAL-SD | $83.0_{\pm 0.3}$ | $90.0_{\pm 0.3}$ | $92.5_{\pm 0.1}$ | $93.5_{\pm 0.1}$ | $\mathbf{94.0}_{\pm 0.1}$ |
| CAL-SDS2 | $83.0_{\pm 0.1}$ | $90.1_{\pm 0.1}$ | $92.7_{\pm 0.2}$ | $93.5_{\pm 0.2}$ | $\mathbf{94.0}_{\pm 0.0}$ |
| AL w/ WS | $82.8_{\pm 0.4}$ | $\mathbf{90.3}_{\pm 0.1}$ | $\mathbf{92.8}_{\pm 0.2}$ | $\mathbf{93.6}_{\pm 0.1}$ | $93.7_{\pm 0.3}$ |
| AL | $72.5_{\pm 2.0}$ | $86.6_{\pm 0.4}$ | $90.1_{\pm 0.4}$ | $91.7_{\pm 0.2}$ | $92.9_{\pm 0.2}$ |

Table 16: CIFAR-10 with FASS

|  | Test Accuracy (%) | | | | |
|---|---|---|---|---|---|
| Method | 10% | 15% | 20% | 25% | 30% |
| CAL-ER | $92.6_{\pm 0.0}$ | $\mathbf{93.9}_{\pm 0.2}$ | $94.3_{\pm 0.1}$ | $\mathbf{94.7}_{\pm 0.1}$ | $94.7_{\pm 0.2}$ |
| CAL-MIR | $92.5_{\pm 0.0}$ | $\mathbf{93.9}_{\pm 0.4}$ | $94.3_{\pm 0.2}$ | $94.4_{\pm 0.2}$ | $94.6_{\pm 0.1}$ |
| CAL-DER | $\mathbf{92.7}_{\pm 0.1}$ | $\mathbf{93.9}_{\pm 0.2}$ | $94.3_{\pm 0.3}$ | $\mathbf{94.7}_{\pm 0.2}$ | $\mathbf{94.9}_{\pm 0.3}$ |
| CAL-SD | $92.6_{\pm 0.1}$ | $93.8_{\pm 0.1}$ | $\mathbf{94.4}_{\pm 0.3}$ | $94.6_{\pm 0.1}$ | $94.7_{\pm 0.1}$ |
| CAL-SDS2 | $92.6_{\pm 0.1}$ | $\mathbf{93.9}_{\pm 0.2}$ | $\mathbf{94.4}_{\pm 0.2}$ | $94.6_{\pm 0.3}$ | $94.7_{\pm 0.2}$ |
| AL w/ WS | $92.5_{\pm 0.1}$ | $93.6_{\pm 0.1}$ | $93.9_{\pm 0.1}$ | $94.1_{\pm 0.1}$ | $94.3_{\pm 0.1}$ |
| AL | $92.5_{\pm 0.2}$ | $93.8_{\pm 0.1}$ | $94.2_{\pm 0.1}$ | $94.6_{\pm 0.2}$ | $94.7_{\pm 0.2}$ |

Table 17: FMNIST with GLISTER

|  | Test Accuracy (%) | | | | |
|---|---|---|---|---|---|
| Method | 10% | 20% | 30% | 40% | 50% |
| CAL-ER | $81.7_{\pm 0.3}$ | $89.2_{\pm 0.2}$ | $91.9_{\pm 0.2}$ | $93.0_{\pm 0.1}$ | $93.3_{\pm 0.1}$ |
| CAL-MIR | $81.6_{\pm 0.3}$ | $89.3_{\pm 0.4}$ | $91.7_{\pm 0.2}$ | $92.9_{\pm 0.1}$ | $93.5_{\pm 0.2}$ |
| CAL-DER | $\mathbf{82.8}_{\pm 0.4}$ | $89.5_{\pm 0.4}$ | $91.7_{\pm 0.4}$ | $92.8_{\pm 0.6}$ | $93.1_{\pm 0.2}$ |
| CAL-SD | $82.5_{\pm 0.3}$ | $\mathbf{89.6}_{\pm 0.2}$ | $92.1_{\pm 0.2}$ | $93.1_{\pm 0.2}$ | $93.8_{\pm 0.1}$ |
| CAL-SDS2 | $81.4_{\pm 0.4}$ | $89.1_{\pm 0.2}$ | $\mathbf{92.1}_{\pm 0.2}$ | $\mathbf{93.2}_{\pm 0.3}$ | $\mathbf{93.9}_{\pm 0.1}$ |
| AL w/ WS | $81.7_{\pm 0.4}$ | $89.3_{\pm 0.4}$ | $\mathbf{92.1}_{\pm 0.3}$ | $93.0_{\pm 0.1}$ | $93.3_{\pm 0.4}$ |
| AL | $81.0_{\pm 0.6}$ | $88.5_{\pm 0.5}$ | $91.5_{\pm 0.3}$ | $93.0_{\pm 0.2}$ | $93.4_{\pm 0.3}$ |

Table 18: CIFAR-10 with GLISTER

### C.3 Effect of Query Size

In this section, we demonstrate that speedups with CAL methods can be realized with different query sizes. We demonstrate this by comparing AL with CAL-SDS2 on CIFAR-10, with three different choices of query sizes. We use entropy based uncertainty sampling as the acquisition function. We observe that CAL-SDS2 achieves comparable performance to standard AL with different query sizes, and the speedup can increase when the query size is reduced.

| Query Size | Method | Test Accuracy (%) | | | | | Factor Speedup | | | | |
|---|---|---|---|---|---|---|---|---|---|---|---|
| | | 10% | 20% | 30% | 40% | 50% | 10% | 20% | 30% | 40% | 50% |
| 5000 | AL | 79.1 | 88.6 | 91.7 | 93.5 | 93.8 | 1.0x | 1.0x | 1.0x | 1.0x | 1.0x |
| | CAL-SDS2 | 79.1 | 89.4 | 92.4 | **94.0** | 94.3 | 1.0x | 1.1x | 1.4x | 1.7x | 2.0x |
| 2500 | AL | 81.9 | 89.6 | 92.4 | 93.5 | 94.1 | 1.0x | 1.0x | 1.0x | 1.0x | 1.0x |
| | CAL-SDS2 | 82.5 | 90.1 | 92.9 | 94.0 | **94.4** | 1.1x | 1.6x | 2.1x | 2.7x | 3.4x |
| 1000 | AL | 84.4 | **90.4** | **93.2** | 93.8 | 94.2 | 1.0x | 1.0x | 1.0x | 1.0x | 1.0x |
| | CAL-SDS2 | **84.3** | 89.8 | 92.5 | 93.2 | 93.5 | 1.6x | 3.1x | 4.2x | 5.3x | 6.4x |

Table 19: Effect of Query Size on CIFAR-10

# D  Additional Details on Single-Cell Cell-Type Identity Classification Dataset

The human cell landscape (HCL) dataset consists of scRNA-seq data for 562,977 cells across 63 cell types represented in 56 human tissues. Each cell type may be present in multiple tissues. The cell type classes are highly imbalanced, with the rarest cell type, human embryonic stem cell, accounting for 0.06 % of the total dataset and the most common, fibroblast, accounting for 6%. The raw data is first normalized for library size and scaled to 10000 reads in total, followed by log transformation. We visualize the dataset using UMAP 15.

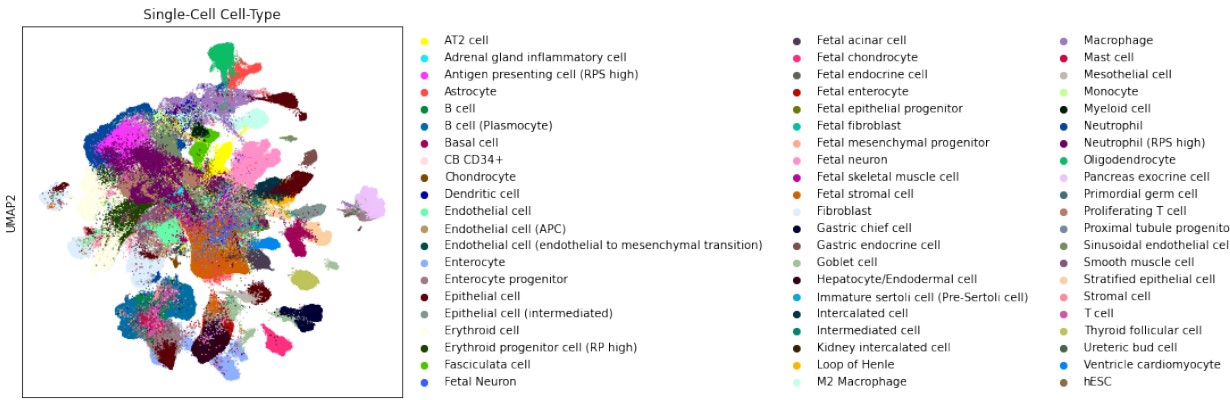

Figure 15: UMAP embedding of single cells in HCL annotated by their cell type.

