# OpenReview forum: "Accelerating Batch Active Learning Using Continual Learning Techniques"
_TMLR — Accepted by TMLR_

### Review · Reviewer_QLBV · 2023-09-20

**Summary Of Contributions:**

This paper proposes a simple strategy for active learning (AL) by integrating replay-based sampling approaches in continual learning (CL) with the classic AL framework, namely continual active learning (CAL), to accelerate the learning in AL. Some experimental results are shown to demonstrate the performance.

**Audience:**

Yes

**Claims And Evidence:**

No

**Requested Changes:**

No specific changes are suggested.

**Strengths And Weaknesses:**

First of all, I am not familiar with the literature of AL and CL. My comments below are based on my understanding of the submission. I'd like to correct them if there is something inaccurate or wrong.

**Strengths:** The proposed method is really simple, and I can see the improvement in learning speed in AL. The paper is written well, and easy to understand.

**Weaknesses:**

1. There is little theoretical analysis on the proposed method, and all the claims are supported by the empirical results.

2. The authors may overclaimed the contributions. Personally, I hardly believe that such batch-based sampling methods, i.e., reducing the number of training samples in a reasonable way for better computational efficiency such as stochastic learning (e.g., SGD), have never been used in AL. Maybe it is not called CL in some methods. If truly nobody has tried this before for better computational efficiency in AL, then I think this submission has merit in the AL field. I am hoping that the other reviewers and AC can better justify their contributions.

---

> ### Author Response · Authors · 2023-09-21
> **Response to Reviewer QLBV**
>
> We are glad that the reviewer found our work easy to follow and well-written. Below we address some of the reviewers concerns:
>
> _Lack of Theory:_
> - Although classical active learning approaches have been extensively studied from a theoretical standpoint, there remains a significant disparity between theory and practice in both batch active learning and continual learning for deep models. The methods presented in this paper are intentionally straightforward, as the purpose of this work is to underscore the significance of a novel problem context, supported by compelling empirical results. We agree with the reviewer that an interesting avenue for future research would involve delving into the theoretical underpinnings of this framework, though it is not within the scope of this work.
>
> _CL to accelerate AL has been explored:_
> - Accelerating active learning is an underexplored area, and the only work that considers reducing the training time of AL models is [4] which is orthogonal to the approach we propose (stated in the Related Works section). We are the firs work, to the best of our knowledge, to demonstrate that active learning can be accelerated by continual learning. We did further google scholar and other web searching and did not find any other work, showing that continual learning can be used to accelerate active learning. Furthermore, existing AL codebases that are widely adopted by the community train from scratch, and do not use any form of experience replay/continual learning [1,2,3,4,5].
>
>
> References:
> - [1] https://github.com/decile-team/distil
> - [2] https://github.com/EfficientTraining/LabelBench
> - [3] https://github.com/zeyademam/active_learning
> - [4] Selection via Proxy: Efficient Data Selection for Deep Learning (https://arxiv.org/abs/1906.11829)
> - [5] https://github.com/ntucllab/libact#libact-pool-based-active-learning-in-python

---

### Review · Reviewer_PnH7 · 2023-09-20

**Summary Of Contributions:**

This paper presents a way of  speeding up batch active learning approaches by leveraging insights from continual learning.
Specifically, they propose to apply continual learning methods to adapting to data from new query rounds whilst also leveraging data from past query rounds instead of restarting model training from scratch.
They show that the continual learning based framework works better than warm restarts both in terms of generalization and speedups.

**Audience:**

Yes

**Broader Impact Concerns:**

No ethical concerns immediately obvious

**Claims And Evidence:**

Yes

**Requested Changes:**

Please see weaknesses above as they also contain requests for clarification

**Strengths And Weaknesses:**

# Strengths #
- the paper introduces a compatible marriage of two pre-existing fields
- the paper conducts quite comprehensive experiments to validate their approach
- paper is clearly written, with the experimental methodology capturing the important questions that the framework poses

# Weaknesses #
There are a few things that are unclear from the paper that I would like resolved.
1. Algorithms 1 and 3 return a set of labeled examples $\mathcal{L}$ -- are the results reported in the paper the final results obtained from $\theta_{T}$ -- the last iterate in the respective procedures, or are they results obtained from  :

             (i) creating a new random initialization $\theta$

             (ii) training the randomly initialized model once on the final $\mathcal{L}$ and then reporting it's final performance ?

2. On page 6 -- section Baselines, the paper states *Both models retrain on the entire dataset at each query round*  -- with respect to the AL methods.  Since a big part of the paper is the speedup, it seems to me that this detail would mean that the other methods always trivially result in speeds up due to using only $m^{(h)}$ samples from the history (granted these $m^{(h)}$ samples are chosen smartly) .  I would therefore contend that the CL-ER method, should rather be considered a baseline AL method (AL-WS-Random Subsample) since it's AL with randomly chosen batch from the history in order tot reduce compute cost.

3.  Some results come without significance testing even though the error bars quite overlap. Could you report the p-values for say Figure 5. It's unclear if the effects here should be taken on face value

---

> ### Author Response · Authors · 2023-09-28
> **Response to Reviewer PnH7**
>
> We thank the reviewer for providing a high quality review and for finding our work clearly written and our experiments thorough.
>
> On Algorithms 1 and 3:
> - The reviewer is absolutely correct in pointing out this lack of clarity. The results reported in the paper are the final results obtained from a model that has been trained on the final $\mathcal{L}$. In Algorithm 1, the model is trained from scratch and in Algorhm 3, the ContinualTrain subroutine is used. We will modify both Algorithms 1 and 3 to include the training of $\theta$ after the final round of AL/CAL and return $\theta_{T}$ in addition to $\mathcal{L}$ for more clarity.
>
> On CAL-ER as baseline:
> - We agree with the reviewer in that CAL-ER could also be referred to as AL-WS-Random Subsample; they are two sides of the same coin. However, Experience Replay (ER) is used in all continual learning works as a baseline but we are not aware of any active learning works which consider training the model on a subset of the labeled pool as a baseline (even in the few works that do explore both AL and AL w/ WS). Therefore, in order to be as consistent as possible with both the CL and the AL literature, we consider ER a CAL technique. Either way, as demonstrated in Figure~1, the other CAL algorithms tend to be more performant than CAL-ER so our work demonstrates that there is a benefit to using better continual learning techniques.
>
> Significance of the Robustness Results:
> - Our motivation for performing the test of robustness is to check whether the speedups from the CAL procedures have a detrimental impact on the robustness of the trained models. We agree that there are overlaps as shown in Figure~5, however, this also means that on an average over 19 robustness tests, CAL doesn’t hurt the generalization when compared to the baseline AL.
> We went ahead and did a pairwise p-value estimation on each of the 19 tests each CAL method with the baseline AL. The p-values are all listed in the table below.
> - In most cases, there is no statistically significant decline in performance of a CAL model when compared to standard AL. Thus, robustness is preserved in these cases.
> - For CAL-DER compared to baseline AL, we observe only for 3 tests p-value < 0.05, and on those 3, AL outperforms CAL-DER on 2 of them.
> - For CAL-SDS2 compared to baseline AL we observe only for 6 tests p-value < 0.05, and on those 6, CAL-SDS2 outperforms AL on 4 of them.
>
>
>
> In the table below, we report the p-value estimates between each CAL method compared to standard AL. We highlight statistically significant cases (p value < .05) and indicate an improvement in performance with (+) and a decline in performance with (-). We will add the above this to the supplementary section of the paper.
>
>
> |             | saturate | impulse_noise | defocus_blur |    contrast   |  frost | speckle_noise | pixelate | zoom_blur | elastic_transform | spatter |  snow  |   fog  | gaussian_noise |   brightness  | gaussian_blur |  motion_blur  |   shot_noise  | jpeg_compression | glass_blur |
> |-------------|:--------:|:-------------:|:------------:|:-------------:|:------:|:-------------:|:--------:|:---------:|:-----------------:|:-------:|:------:|:------:|:--------------:|:-------------:|:-------------:|:-------------:|:-------------:|:----------------:|:----------:|
> | CAL-SDS2/AL |  0.7961  |     0.5374    |    0.3178    | **0.0250(-)** | 0.2778 | **0.0017(+)** |  0.5636  |   0.3910  |       0.5176      |  0.7800 | 0.1976 | 0.7142 |  **0.0013(+)** | **0.0474(-)** |     0.2997    |     0.7135    | **0.0004(+)** |   **0.0000(+)**  |   0.1182   |
> |  CAL-DER/AL |  0.2118  |     0.7029    |    0.1561    | **0.0256(-)** | 0.2147 |     0.7409    |  0.4661  |   0.4279  |       0.9975      |  0.4310 | 0.3747 | 0.2781 |     0.7836     | **0.0341(-)** |     0.2605    |     0.6263    |     0.7311    |   **0.0016(+)**  |   0.7633   |
> |  CAL-SD/AL  |  0.4516  |     0.7134    |    0.7854    |     0.1020    | 0.9548 |     0.9739    |  0.3913  |   0.7238  |       0.9154      |  0.5843 | 0.5455 | 0.4512 |     0.8056     | **0.0091(-)** |     0.8658    |     0.6801    |     0.9091    |   **0.0286(+)**  |   0.5807   |
> |  CAL-MIR/AL |  0.9104  |     0.5692    |    0.2794    |     0.3023    | 0.6639 |     0.9922    |  0.2521  |   0.5567  |       0.1295      |  0.9401 | 0.8905 | 0.2534 |     0.7485     |     0.2042    |     0.5867    | **0.0368(+)** |     0.8920    |      0.1080      |   0.1047   |
> |  CAL-ER/AL  |  0.3957  |     0.3209    |    0.2477    | **0.0335(+)** | 0.5793 |     0.9873    |  0.2243  |   0.2824  |       0.5482      |  0.1504 | 0.7823 | 0.4715 |     0.3514     | **0.0309(-)** |     0.2969    |     0.6474    |     0.7252    |   **0.0110(+)**  |   0.2820   |

---

### Review · Reviewer_RHXd · 2023-09-21

**Summary Of Contributions:**

Overall, the paper explores an important problem and presents an intriguing idea in the area of batch active learning, specifically focusing on leveraging continual learning techniques for acceleration.

**Audience:**

Yes

**Claims And Evidence:**

Yes

**Requested Changes:**

My main concerns are as follows:
Lack of Innovation: While the paper addresses an important problem and employs continual learning techniques, it falls short in terms of introducing substantial innovative contributions or novel algorithms. It would benefit from proposing new concepts or extensions to existing methods to differentiate itself from prior works. The contribution of Scaled Distillation is not very impressive, in my opinion, it doesn't have any fundamental difference from DER.
Comparatively Weak Methodology: The proposed approach (SD and SD2) is deemed less effective than existing methods in the field. It is crucial to clearly highlight the limitations and shortcomings of the proposed method and provide a comparative analysis that demonstrates its inferior performance compared to state-of-the-art alternatives.
Limited Performance Analysis: The paper could provide more in-depth analysis and discussion of the experimental results. By thoroughly examining the performance metrics and presenting insightful interpretations, it would offer a better understanding of the current limitations of existing methods and areas for potential improvement.

**Strengths And Weaknesses:**

Strengths:
Importance of the Research Problem: The paper addresses a significant problem in the field of active learning, which is the time cost. This is a relevant and challenging issue that has practical implications in various domains.
Neat Ideas: The use of continual learning techniques to accelerate batch active learning is an interesting concept and shows potential for advancements in the field. It brings together two distinct research areas, which can contribute to the development of more efficient and effective active learning methods.
Experimental Validation on Multiple Datasets: One commendable aspect of the paper is the thorough evaluation of the proposed approach on multiple datasets. Conducting experiments on diverse datasets strengthens the validity and applicability of the findings.

Weaknesses:
Lack of Innovation: While the paper addresses an important problem and employs continual learning techniques, it falls short in terms of introducing substantial innovative contributions or novel algorithms. It would benefit from proposing new concepts or extensions to existing methods to differentiate itself from prior works. The contribution of Scaled Distillation is not very impressive, in my opinion, it doesn't have any fundamental difference from DER.
Comparatively Weak Methodology: The proposed approach (SD and SD2) is deemed less effective than existing methods in the field. It is crucial to clearly highlight the limitations and shortcomings of the proposed method and provide a comparative analysis that demonstrates its inferior performance compared to state-of-the-art alternatives.
Limited Performance Analysis: The paper could provide more in-depth analysis and discussion of the experimental results. By thoroughly examining the performance metrics and presenting insightful interpretations, it would offer a better understanding of the current limitations of existing methods and areas for potential improvement.
In conclusion, the paper addresses an important problem and presents an interesting idea by exploring the use of continual learning techniques for accelerating batch active learning. However, it lacks sufficient innovation, falls short in terms of method effectiveness compared to existing approaches, and requires stronger analysis and discussion of experimental performance. To enhance the paper's contribution and clarify its limitations, further advancements in the proposed method and more extensive evaluation would strengthen its overall scientific value.

---

> ### Author Response · Authors · 2023-09-28
> **Response to Reviewer RHXd**
>
> Thank you for your comprehensive review! We are glad that you found the problem setting interesting and the experimental results compelling.
>
> Lack of Innovation:
> - Accelerating active learning is a very underexplored area. Thus, the question that our work seeks to answer is “can active learning be accelerated with continual learning without affecting downstream performance?” by establishing the CAL framework. Thus, as we state in our paper, the aim of this work is not to champion a single method, but rather to showcase an assortment of approaches in the CAL paradigm that achieve different performance/speedup trade-offs. We agree with the reviewer that novel CL algorithms specifically tailored towards speeding up AL would be a very interesting direction to pursue as we specify in the Future Work section.
>
> On limitations of the proposed methods:
> - Given our very diverse suite of evaluation – vision/NLP/genomics, one may consider the lack of a clear winner as a limitation. However, as we mention in the paper, the aim of this work is not to champion a single method, but rather to showcase an assortment of approaches in the CAL paradigm that achieve different performance/speedup trade-offs. Let us know if there are any specific claims that the reviewer feels are substantiated and we can make additional changes accordingly.
>
> Limited Performance Analysis:
> - We respectfully disagree with the reviewer on the point that our experimental analysis was not sufficiently in-depth. Not only do we test on a diverse set of datasets spanning multiple types of modalities and architectures, but we also report much more than just accuracy. We assess the robustness of our resulting models (Figure 5 and Appendix A3 for breakup with respect to each corruption), analyze the correlation statistics (Figure 6), and also provide a sensitivity analysis for our proposed methods, with respect to the query size (Appendix C3) as well as the hyperparameters (Appendix A6, Table 9 and Table 10). However, if there is a specific set of analyses that the reviewer feels is missing from our submission, we would be happy to discuss the possibility of including them in the next version of the paper.
>
> Please let us know if there are any further questions or clarifications you would like from us!

---

### Comment · Action_Editors · 2023-09-29
**High-level questions**

Dear authors,

Thanks for submitting to TMLR. I roughly read the reviews and had a quick look at the paper. I would agree that this paper addresses an important and unexplored problem in modern (deep) active learning. Meanwhile, I have some high-level questions. I am not sure if these questions are discussed in the paper. Here are just my initial thoughts after a quick review. It would be great if the authors could provide some clarifications.

- Why does standard AL on neural networks with warm starting fail? In active learning, we generally assume that the data comes from one underlying distribution, while in continual learning, forgetting happens when we face a *distribution shift*, right? Why does active learning have a similar problem?

- Does the choice of a continual learning approach matter? We know that there are many ways to address forgetting, such as memory-based, replay, regularization, etc. Is it important to choose a particular approach in active learning?

---

> ### Author Response · Authors · 2023-10-02
> **Response to High Level Questions**
>
> Thank you for your comments! We answer both questions below:
>
> Why is there forgetting in standard active learning with warm starting?
> - In the paper, we claim that the primary issue with standard AL with warm starting is its inefficiency. AL with warm-start still requires making passes through the full labeled set, so the training time increases as the labeled pool increases (even if the number of training steps required is slightly reduced). Furthermore, there is some empirical evidence that suggests that warm-start neural network training can hamper generalization in general though there is no unanimous consensus [1,2].
> - Forgetting occurs if we solely train the model on newly labeled data, which is distinct from standard AL with warm starting which trains on the entire labeled pool. In AL, we gradually explore the entire data distribution through a series of query rounds conducted on a pool of examples. Initially, in the first round, the seed set is typically chosen uniformly at random. However, for subsequent rounds, the selection of query examples is conditioned on the currently trained model. Hence, there is a distribution shift across the successive query rounds which we observe in Figure 2.
>
> Does the choice of a continual learning approach matter?
>
> - The CAL framework is general, so any continual learning approach can be used to accelerate AL. However, memory-based and regularization based CL approaches impose restrictions that are not necessary in the CAL setting. We discuss these below:
>
> - Regularization based approaches: Regularization based approaches [3] assume that it is not possible to retain any examples from the history. This assumption is not applicable to the CAL setting, since the labeled pool is fully accessible in AL. When the setting does not preclude using examples from the history, regularization based approaches tend to significantly underperform replay/memory based CL approaches [4,5].
> - Memory based approaches: Memory based CL approaches [6] focus on developing approaches that determine which samples from the history should be retained. This is useful if it is the case that the history is too large to store in memory, as is typically the case in CL when the stream of data is of indeterminate length. However, in pool-based AL the full unlabeled and labeled pools are already stored, so employing a memory-based approach would unnecessarily discard examples.
>
> - We therefore opt to focus on replay based CL techniques as these approaches do not impose any restrictions that are not applicable to the CAL setting and would unnecessarily reduce performance. This is briefly mentioned in the related works section. However, we will include a separate discussion in the next version of the paper, as we believe this to be useful to the audience.
>
> Please let us know if there are any remaining questions.
>
> References:
> - [1] Effective Evaluation of Deep Active Learning on Image Classification Tasks (https://arxiv.org/pdf/2106.15324.pdf )
> - [2] On Warm-Starting Neural Network Training (https://proceedings.neurips.cc/paper/2020/file/288cd2567953f06e460a33951f55daaf-Paper.pdf )
> - [3] Overcoming catastrophic forgetting in neural networks (https://arxiv.org/pdf/1612.00796.pdf)
> - [4] Dark Experience for General Continual Learning: a Strong, Simple Baseline (https://proceedings.neurips.cc/paper/2020/file/b704ea2c39778f07c617f6b7ce480e9e-Paper.pdf )
> - [5] GDumb: A Simple Approach that Questions Our Progress in Continual Learning (https://www.ecva.net/papers/eccv_2020/papers_ECCV/papers/123470511.pdf )
> - [6] Gradient based sample selection for online continual learning (https://arxiv.org/abs/1903.08671 )

---

> > ### Comment · Action_Editors · 2023-10-04
> >
> > Thank you very much! These make sense to me. I will read the paper in detail later and discuss it with the reviewers.

---

### Decision · Action_Editor_gu9A · 2023-10-25

**Recommendation:** Accept with minor revision

**Comment:**

This paper considered batch active learning, with a particular focus on using continual learning techniques to speed up the training process in modern deep active learning.

I think this paper considered an unexplored but important problem in active learning -- using continual learning without training from scratch. In fact, this AE has also been considering this issue for a while and is quite happy to see the integration of continual learning to address these training problems.

At the same time, some reviewers raised concerns about the manuscripts. Here are the main points and my suggestions.

- [Justifications] Some reviewers think that this paper could be much stronger if a systematic analysis was done to understand the choice of continual learning methods. I noticed that the authors updated one paragraph to reflect this. For the final version, I would suggest a thorough discussion. For example, a comparative table to illustrate the pros and cons of different continual learning approaches in active learning.

- [Distribution shift/forgetting/theoretical support] Reviewer QLBV thought of a lack of rigorous analysis, and I wondered about the source of forgetting. Upon reflection, I would think that the theoretical support might be explained by a distribution shift viewpoint. I would recommend two related papers [1-2]. In particular, paper [1] considered a dynamic update (analogous to continual learning) in active learning, based on the distribution shift in paper [2]. I would like to suggest a brief discussion of the theoretical possibilities.

Based on these, I would recommend an accept with a minor revision.

References:

[1] Active Learning Using Discrepancy. https://realworldml.github.io/files/cr/7_cui_paper.pdf

[2] Deep Active Learning: Unified and Principled Method for Query and Training. AISTATS, 2020.

**Audience:**

Yes.

**Claims And Evidence:**

Yes. Some parts need minor revisions.